# Beyond Text-to-SQL: Can LLMs Really Debug Enterprise ETL SQL?

**Jing Ye** [1]   **Yiwen Duan** [2]   **Yonghong Yu** [2]   **Victor Ma** [1]   **Yang Gao** [2]   **Xing Chen** [2]

## Abstract

SQL is central to enterprise data engineering, yet generating fully correct SQL code in a single attempt remains difficult—even for experienced developers and advanced Text-to-SQL LLMs—often requiring multiple debugging iterations. We introduce **Squirrel Benchmark**, the first benchmark for enterprise-level SQL reasoning and debugging. Our benchmark is built upon two key innovations: (1) an **automated construction workflow** that employs reverse engineering to systematically inject realistic bugs into large-scale SQL code, enabling scalable and diverse benchmark generation; and (2) an **execution-free evaluation framework** tailored for enterprise settings, providing fast, accurate, and resource-efficient assessment. Squirrel Benchmark comprises 469 Squirrel-Syntax queries featuring syntax errors with explicit error messages, and 516 Squirrel-Semantic queries targeting semantic errors where codes fails to meet user intent. The queries are highly complex, averaging over 140 lines, and featuring deep and wide abstract syntax trees (average width $> 11$, depth $> 8.7$). Evaluation of nearly 30 LLMs reveals a substantial performance gap: the best-performing model, Claude-4-Sonnet, achieves only 36.46% accuracy on Squirrel-Syntax and 32.17% on Squirrel-Semantic, while most models score below 20%. We further explore four solution strategies, identify key challenges, and outline promising directions for enterprise SQL debugging with LLMs.

## 1 Introduction

Databases are a cornerstone of modern data infrastructure, powering critical applications across finance, web services,

and scientific computing. Structured Query Language (SQL) remains the predominant interface for human–data interaction, enabling large-scale extraction, transformation, and loading (ETL) workflows (Chamberlin & Boyce, 1974; Armbrust et al., 2015). Recent research on Text-to-SQL large language models (LLMs) has sought to help analysts automate routine queries, streamline data workflows, and support advanced business intelligence (Zhong et al., 2018; Yu et al., 2018; Li et al., 2025).

Enterprise SQL code is often lengthy, complex, and deeply nested, making it extremely challenging for both experienced developers and Text-to-SQL LLMs to generate correct code in a single attempt (Lei et al., 2025). Instead, success typically requires multi-step reasoning and iterative debugging. As shown in Figure 1, debugging generally involves localizing errors, analyzing their causes, consulting schema definitions, applying targeted modifications, and re-running lint checks to verify whether requirements are satisfied—usually repeating this loop multiple times. Unfortunately, LLMs struggle with this iterative correction process. They frequently fall into anti-patterns such as repeating identical actions without meaningful follow-up, which leads to wasted effort when an initial correction fails (Bouzenia & Pradel, 2025; Laban et al., 2026).

To bridge this gap, we propose moving beyond Text-to-SQL generation and shifting the focus to a model's ability to iteratively debug and self-correct. We introduce Squirrel Benchmark, a benchmark for evaluating LLMs on enterprise-scale SQL debugging. Our construction pipeline uses an automated reverse-engineering framework to synthesize realistic, reproducible tasks. This approach minimizes human effort while ensuring high-quality benchmark generation, also providing a foundation for synthetic training data. Furthermore, we design an execution-free evaluation framework tailored to enterprise SQL scenarios. Squirrel Benchmark offers a practical reference point for selecting SQL-focused LLMs in industry. The benchmark comprises 469 Squirrel-Syntax tasks (syntax errors with explicit error messages) and 516 Squirrel-Semantic tasks (semantic errors in which the SQL output does not match the user's requirement). SQL programs in our benchmark are highly complex, averaging over 140 lines ($> 420$ tokens), with ASTs of width $> 11$ and depth $> 8.7$, and incorporating

---

[1]Independent Researcher [2]Bytedance Inc., Beijing, China. Correspondence to: Xing Chen <chenxing.xc@bytedance.com>.

*Proceedings of the $43^{rd}$ International Conference on Machine Learning*, Seoul, South Korea. PMLR 306, 2026. Copyright 2026 by the author(s).

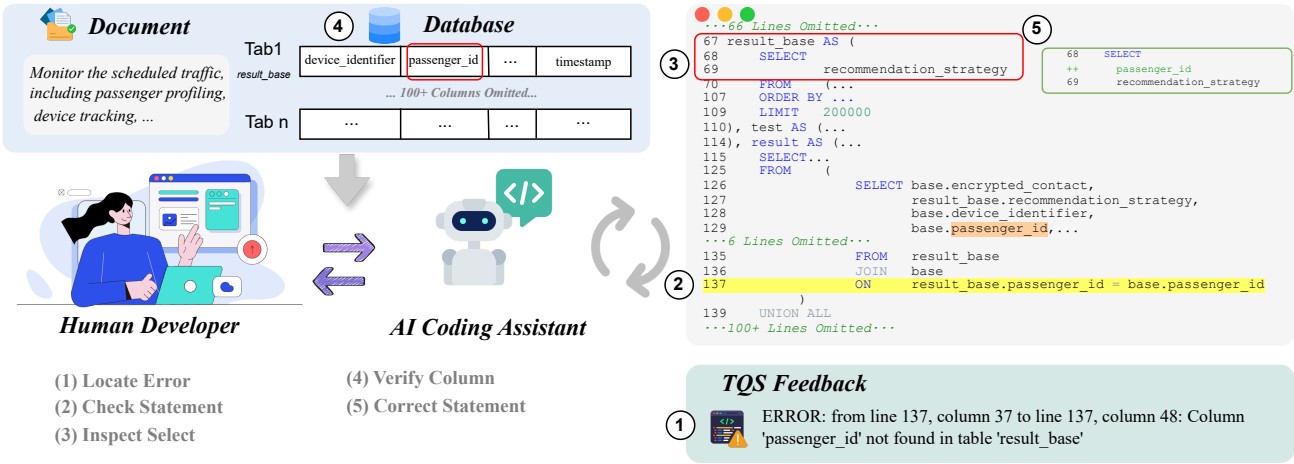

*Figure 1.* Squirrel Benchmark evaluates LLMs on real-world enterprise-level SQL debugging workflows. It involves multi-step reasoning and actions, including understanding requirements and schemas, diagnosing error messages, and iteratively refining scripts through cycles of reasoning and debugging.

over 15 functions per script.

Our evaluation on Squirrel Benchmark indicates significant room for improvement in deploying LLMs within SQL-SWE workflows. Extensive experiments show that even state-of-the-art LLMs struggle: Claude-4-Sonnet achieves only 36.46% success on Squirrel-Syntax and 33.17% on Squirrel-Semantic, while most models fail to reach 20%. These results underscore the difficulty of enterprise SQL debugging and highlight substantial room for improvement. To address this gap, we systematically explore four potential solution strategies and conduct comprehensive experiments to assess their effectiveness. Our results not only illuminate the challenges LLMs face in SQL debugging but also offer insights into strategies to improve performance. Moreover, Squirrel Benchmark exhibits a strong correlation with real-world debugging outcomes, establishing it as a reliable benchmark for aligning models with industrial applications. In summary, this work makes the following contributions:

- We propose an automated reverse-engineering workflow for constructing high-quality SQL debugging benchmarks, which can also be adapted to synthesize realistic training data.

- We present Squirrel Benchmark, a large-scale benchmark comprising 469 syntax and 516 semantic tasks, designed to capture the complexity, diversity, and practicality of enterprise SQL development.

- We conduct a comprehensive evaluation of nearly 30 open-source and proprietary LLMs, showing that even the state-of-the-art LLMs face substantial challenges.

- We introduce three SFT and an agent method as baselines, offering a novel and efficient pathway for further studies.

**Conflict of Interest Disclosure** Some of the authors are employees of ByteDance. The evaluated models include systems developed by a variety of organizations, including organizations affiliated with the authors' employer. All models were used solely for research and evaluation purposes under their applicable terms of use. The analyses and conclusions presented in this paper are independent academic findings and do not necessarily represent the views of any model developer or provider.

## 2 Preliminary

### 2.1 Task Definition

SQL debugging is a fundamental but underexplored problem in data development. Existing Text-to-SQL research primarily focuses on translating natural language to SQL queries, but real-world scenarios often involve correcting issues in SQL scripts. The goal of SQL debugging is to automatically repair buggy SQL scripts while preserving the user's intent. This task begins with a buggy SQL query $(b)$, accompanied by auxiliary context $\mathcal{C}$ (e.g., error messages or natural-language intent descriptions) and the database schema $(\sigma)$. The objective is to generate a corrected SQL $(\hat{q})$:

$$\hat{q} = f_\theta(\mathcal{C}, \sigma, b) \tag{1}$$

where $\hat{q}$ is syntax correct and faithful to the intent encoded in $(\mathcal{C}, b, \sigma)$.

We categorize bugs into two primary types: **(I) Syntactic errors.** $b$ is non-executable. Here, $\mathcal{C}$ is the error message $\mathcal{E}$, and the goal is to produce an executable repair while preserving its intended semantics. **(II) Semantic errors.** $b$ executes successfully but fails to meet the user's requirements. In this case, $\mathcal{C}$ is a natural language specification $\mathcal{R}$,

and the task is to modify $\hat{q}$ to satisfy $\mathcal{R}$. By covering both types, Squirrel Benchmark unifies execution repair with intent comprehension, offering a challenging and realistic benchmark for SQL debugging.

## 2.2 Challenges

Despite its practical importance, SQL debugging introduces several unique challenges that are not sufficiently addressed in existing SWE research.

**Challenge 1: Lack of Enterprise-level SQL Scripts.** Industrial SQL workloads, such as ETL workflows and scheduled analytical jobs, are typically *long*, *complex*, and *schema-heavy*. Scripts can span hundreds of lines, involve deeply nested subqueries and multi-way joins. and reference dozens of tables and columns. This level of intricacy significantly amplifies the challenge for LLMs. In contrast, most existing Text-to-SQL (Li et al., 2024) and SQL-debugging (Li et al., 2026) benchmarks focus on short, relatively simple queries that are far removed from the scale and complexity of enterprise environments. Unfortunately, such industrial-grade SQL scripts are rarely available in the open-source community, resulting in a pronounced mismatch between academic benchmarks and real-world needs.

> **Contribution 1:**
>
> To address this gap, we introduce a large-scale, enterprise-level benchmark that captures the complexity of real-world ETL and analytical workloads (Section 3.1).

**Challenge 2: Lack of a Comprehensive Bug Taxonomy.** SQL bugs are heterogeneous: some manifest as execution failures (syntax errors), while others silently yield incorrect results (semantic errors). Although recent benchmarks such as BIRD-Critic (Li et al., 2026) have advanced debugging evaluation, they lack a systematic taxonomy of SQL-specific bug types and their prevalence. Without such categorization, it is difficult to understand where models struggle most and how to target improvements effectively. A comprehensive analysis of SQL bug categories is therefore crucial, not only for benchmarking but also for guiding the design of future bug-fixing models.

> **Contribution 2:**
>
> We develop a hierarchical taxonomy of SQL bug types derived from an extensive analysis of real-world errors. This provides a structured framework for fine-grained evaluation (Section 3.2).

**Challenge 3: Lack of Reliable and Comprehensive SQL debugging Benchmark.** High-quality benchmarks for SQL Debugging are scarce. Manually curated datasets are costly to produce and prone to evaluation leakage if models memorize solutions from public templates or repositories. Existing resources often lack diversity, realistic bug patterns, and coverage of enterprise-scale scripts, limiting their usefulness for robust model evaluation. Building a reliable, large-scale benchmark that is both comprehensive and faithful to real-world workflows is therefore a significant challenge.

> **Contribution 3:**
>
> We introduce an automated pipeline for synthesizing and validating SQL bug-fixing examples, ensuring scalability, diversity, and resistance to data leakage (Section 3.3).

## 3 Squirrel Benchmark Construction

Figure 2 shows the automated benchmark construction pipeline. It comprises four stages: (1) enterprise-level SQL script generation (Section 3.1), (2) SQL bug taxonomy design (Section 3.2), (3) issue SQL construction via reverse engineering (Section 3.3), and (4) validation and analysis (Section 3.4). Section 3.5 further introduces an efficient execution-free evaluation methodology. Section 3.5 presents an execution-free evaluation methodology. Examples and prompts are detailed in Appendix J and H.

### 3.1 Enterprise-level SQL Scripts Generation

Because enterprise SQL scripts are proprietary and rarely accessible, we synthesize realistic, high-quality enterprise SQL.

**Seed Enterprise SQL Curation.** We curate high-quality SQL scripts $q$ along with corresponding table definitions $\sigma$ from real-world enterprise applications. To ensure that queries are non-trivial and representative of practical workloads, we filter scripts that fall below a complexity threshold $\tau$. Complexity is quantified via a composite metric:

$$\mathcal{C}(q) = \alpha\big(D_{\text{AST}}(q) + W_{\text{AST}}(q)\big) + \beta L(q) \qquad (2)$$

where $D_{\text{AST}}$, $W_{\text{AST}}$, and $L(q)$ denote AST depth, AST width, and code length, respectively.

For each retained SQL script, we utilize an LLM to abstract its business domain ($d$), intention ($I$), and descriptive scenario ($S$). All scenarios are aggregated into a Scenarios Library, denoted as $\mathcal{D}_{\text{domain}} = \{d\}$. The resulting seed dataset is then defined as:

$$\mathcal{D}_{\text{seed}} = \{(q_i, \sigma_i, d_i, I_i, S_i, \text{AST}(q_i)) | q_i \in \mathcal{Q}_s, \mathcal{C}(q_i) > \tau\},$$
$$(3)$$

where $\mathcal{Q}_s$ denotes the candidate SQL pool.

The final seed corpus contains 1,000+ SQL scripts spanning 26 business scenarios, averaging over 120 lines with AST depth $> 8$ and width $> 12$. Each script is rigorously validated to be bug-free, resulting in a corpus that accurately captures both the structural complexity and semantic diversity of enterprise SQL.

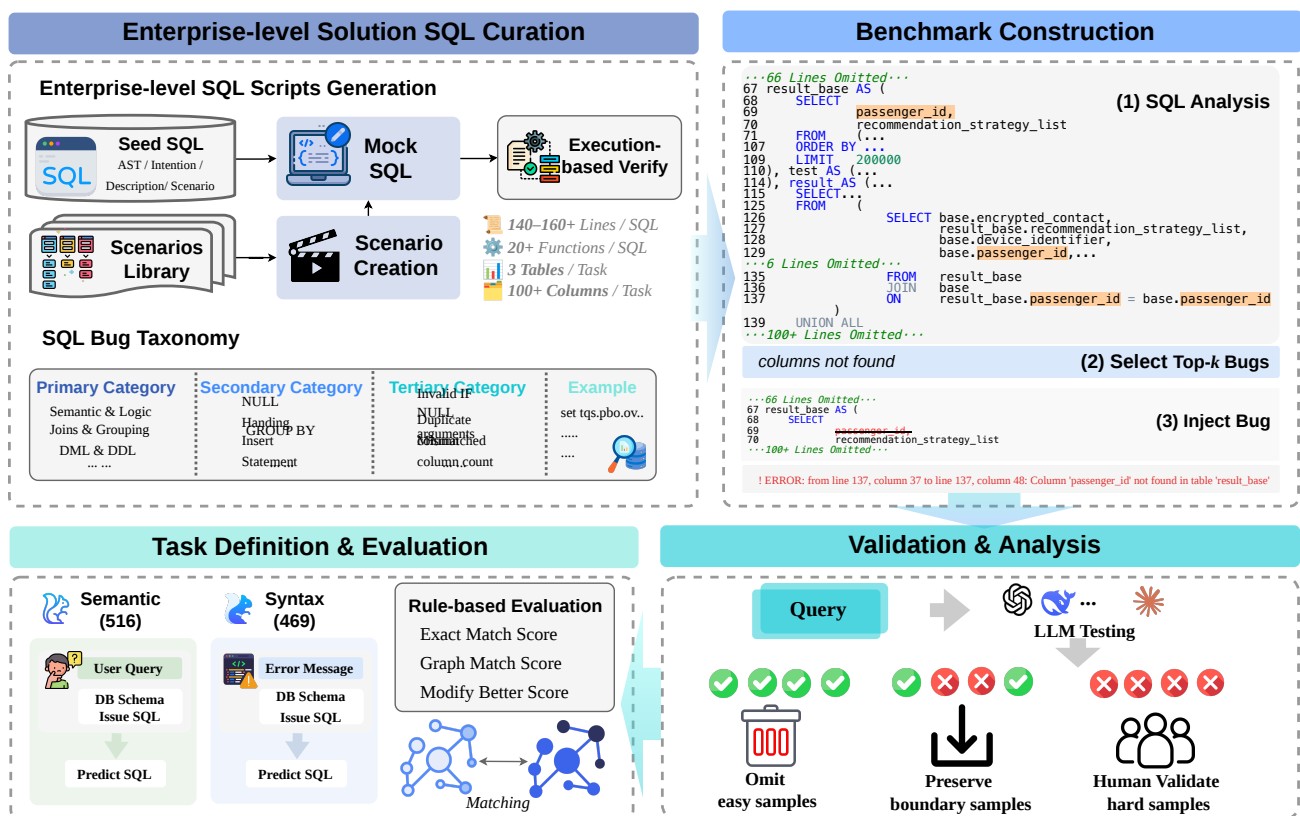

*Figure 2.* Overview of the Squirrel Benchmark construction and evaluation pipeline. Benchmark construction consists of 4 main stages: (1) **Enterprise-level SQL Script Generation**, (2) **SQL bug taxonomy Design**, (3) **Issue SQL Construction via reverse engineering**, and (4) **Validation and Analysis**. This pipeline ensures diversity, realism, and rigorous evaluation of the SQL Debugging task.

**Solution SQL Synthesis.** To expand coverage across domains and code structures, we synthesize new SQL scripts using the seed corpus and the Scenarios Library:

1. *Seed Sampling.* Select $(q_i, \sigma_i, d_i, I_i, S_i, \mathrm{AST}(q_i)) \in \mathcal{D}_{\mathrm{seed}}$ and a target domain $d_t \in \mathcal{D}_{\mathrm{domain}}$.

2. *Scenario Creation.* Conditioned on $d_t$, the LLM generates a new scenario description $\mathcal{S}_t$ together with schema definitions $\sigma_t$, following the structure of the seed corpus.

3. *SQL Synthesis.* Given $(I_i, S_i, \mathrm{AST}(q_i), \mathcal{S}_t, \sigma_t)$, the LLM generates a new SQL script $q_t$ that preserves the complexity of the seed SQL scripts while adapting to the new schema and scenario. This ensures that synthesized queries remain realistic, non-trivial, and representative of enterprise workloads.

4. *Execution-based Validation.* To ensure the correction, each candidate $q_t$ is validated via execution. Specifically, $\sigma_t$ is instantiated to construct a fake test database, $q_t$ is executed, and only queries that successfully execute are retained:

$$\mathcal{Q}_{\mathrm{gt}} = \big\{ (q_t, \sigma_t) \mid \mathrm{exec}(q_t, \sigma_t) == \mathrm{passed} \big\} \quad (4)$$

This synthesis pipeline ensures that the final SQL dataset exhibits (i) enterprise-grade complexity, (ii) broad domain coverage via controlled scenario transfer, and (iii) guaranteed execution correctness.

### 3.2 SQL Bug Taxonomy

We construct an SQL bug taxonomy by manually annotating 268 erroneous SQL scripts collected from real-world production logs. Each bug is classified according to a three-level hierarchical error type: (i) *macro categories* (e.g., DML, DDL, semantic, and logic), (ii) *construct-specific subcategories* (e.g., `INSERT` statements), and (iii) *atomic faults* (e.g., mismatched column counts). This taxonomy organizes common failure patterns and forms a bug library of realistic error templates. The library underpins our controlled bug-injection process (Section 3.3), ensuring that Squirrel Benchmark captures authentic SQL error modes. Table 4 and 3 report the distribution of bug types.

### 3.3 Issue SQL Construction

We construct issue SQL queries through reverse engineering, transforming correct SQL scripts into buggy versions. The process is guided by three principles: structural awareness,

taxonomy-guided selection, and minimal-change injection, ensuring that the generated bugs are both realistic and diagnostically useful.

**Step 1: Structural Profiling and Taxonomy-Guided Selection** For each ground-truth SQL $q_{gt}$, we first analyze its structural and semantic profile, including the AST, function patterns, and clause usage. Based on this profile, we then select the top-$k$ candidate bug types from our hierarchical SQL bug taxonomy. This approach ensures that the injected errors are well-suited to the given SQL while providing broad coverage of real-world error scenarios.

**Step 2: Minimal Change-Based Bug Injection.** Each injected bug represents the smallest possible modification that induces the targeted error type. This principle preserves maximal similarity between the buggy SQL $b$ and its reference $q_{gt}$, isolating the error signal and reducing confounding factors. As a result, evaluating whether a model can localize and repair the fault becomes both precise and interpretable.

### 3.4 Validation and Analysis

We validate Squirrel Benchmark via a model-driven *attack–defense* process. The goal is to filter out trivial cases that most models can easily solve, while retaining challenging but solvable instances that better reflect real-world debugging.

**Automated Verification.** We first attack the benchmark by evaluating each generated instance with a diverse set of advanced LLMs (including Qwen3-Coder-32B(Yang et al., 2025), GPT-5(Openai, 2025), DeepSeek-V3.1(DeepSeek, 2025), Claude-4-sonnet(anthropic, 2025), and others). Instances fall into three categories: (i) If the majority of models succeed, the instance is deemed too easy and discarded; (ii) If only a few models succeed, the instance is considered an edge case and retained; (iii) If none of the models succeed, the instance is flagged for manual review. This adversarial filtering ensures that the benchmark emphasizes cases where current models diverge, thereby sharpening its discriminatory power.

**Human Verification.** Instances flagged as potentially unsolvable are subjected to manual inspection by three expert annotators with extensive SQL experience. Following a cross-validation protocol, annotators assess whether the task is logically inferable from the provided context and whether multiple valid solutions exist. Instances that fail to meet these criteria are removed. For cases where multiple correct answers are possible, annotators supplement the benchmark with all valid alternative solutions.

Through this *attack–defense* protocol, Squirrel Benchmark removes trivial cases, yielding a challenging yet solvable testbed.

### 3.5 Evaluation Metrics

The prevailing metrics for SQL debugging are Exact Match (EM) and Execution Accuracy. However, EM is notoriously strict, failing to credit semantically equivalent queries with divergent syntax. Execution Accuracy, while more forgiving, introduces false positives when test databases lack the necessary content to reveal logical errors (Zhan et al., 2025). Direct execution in production also poses practical barriers, being computationally expensive and raising data privacy concerns. To overcome these challenges, we introduce an execution-free evaluation framework based on three metrics (Detailed definitions and formulas are provided in Appendix D.1.2.):

(1) **Exact Match Score (EM)**: This metric assesses strict syntactic correctness by checking for string-level identity between the predicted and reference SQL queries, thereby serving as a baseline for syntactic alignment.

(2) **Graph Match Score (GM)**: This metric evaluates structural and functional equivalence by comparing the optimized abstract syntax tree of the predicted and reference queries, thereby capturing semantic correctness where EM fails.

(3) **Modify Better Score (MB)**: This metric gauges iterative improvement capability by comparing the edit distances from the predicted SQL and the original SQL to the reference, thereby measuring how much closer the refinement is to the target.

## 4 Benchmark Statistics

We present a statistical analysis of Squirrel Benchmark, comparing it with existing SQL datasets in Table 1 and Figure 3. The benchmark emphasizes *complexity* and *realism*, closely reflecting real-world industrial SQL challenges in script structure, error taxonomy, and task diversity.

**Complexity of SQL Scripts.** The SQL scripts in Squirrel Benchmark are not only longer but also structurally more complex, presenting challenges that better mirror real-world enterprise systems. With an average length of $140 - 160$ lines and over 420 tokens, our scripts are an order of magnitude larger than those in BIRD-Critic (which average under 10 lines). This scale directly implies a higher probability of errors and a greater need for models to maintain long-range context and dependency understanding. Additionally, the high number of functions per script (17.34 in Squirrel-Semantic, 21.62 in Squirrel-Syntax) necessitates reasoning across multiple subqueries and nested expressions—a capability that many existing sequence-to-sequence models lack. This scale and functional richness underscore the increased complexity and practical difficulty of the debugging tasks in our benchmark.

*Table 1.* Statistical comparison of Squirrel Benchmark with representative text-to-SQL and SQL debugging benchmarks. The table evaluates benchmarks on scale (# examples), script length (avg. tokens and lines), and structural complexity (avg. function count, AST depth, and width).

| Benchmark | Type | # Test Examples | Length of SQL | | Complex of SQL | | |
| --- | --- | --- | --- | --- | --- | --- | --- |
| | | | # Tok. /SQL | # Line. /SQL | # Func. /SQL | # AST Depth /SQL | # AST Width /SQL |
| Spider 1.0 (Yu et al., 2018) | Text-to-SQL | 2,147 | 18.50 | — | — | — | — |
| Spider 2.0-snow (Lei et al., 2025) | Text-to-SQL | 121 | 154.63 | 56.12 | 14.90 | **11.95** | 9.66 |
| Spider 2.0-lite (Lei et al., 2025) | Text-to-SQL | 256 | 131.79 | 49.84 | 13.65 | **11.97** | 10.05 |
| BIRD (Li et al., 2024) | Text-to-SQL | 1,789 | 30.90 | — | — | — | — |
| BIRD-Critic-open (Li et al., 2026) | SQL debugging | 600 | 49.18 | 9.73 | 4.30 | 8.03 | 6.01 |
| BIRD-Critic-postgresql (Li et al., 2026) | SQL debugging | 530 | 51.44 | 6.92 | 4.78 | 8.25 | 6.34 |
| BIRD-Critic-flash (Li et al., 2026) | SQL debugging | 200 | 34.53 | 2.84 | 4.06 | 7.85 | 5.20 |
| Squirrel-Syntax | SQL debugging | 469 | **496.90** | **163.69** | **21.62** | 8.93 | **11.69** |
| Squirrel-Semantic | SQL debugging | 516 | **425.93** | **141.58** | **17.34** | 8.75 | **11.12** |

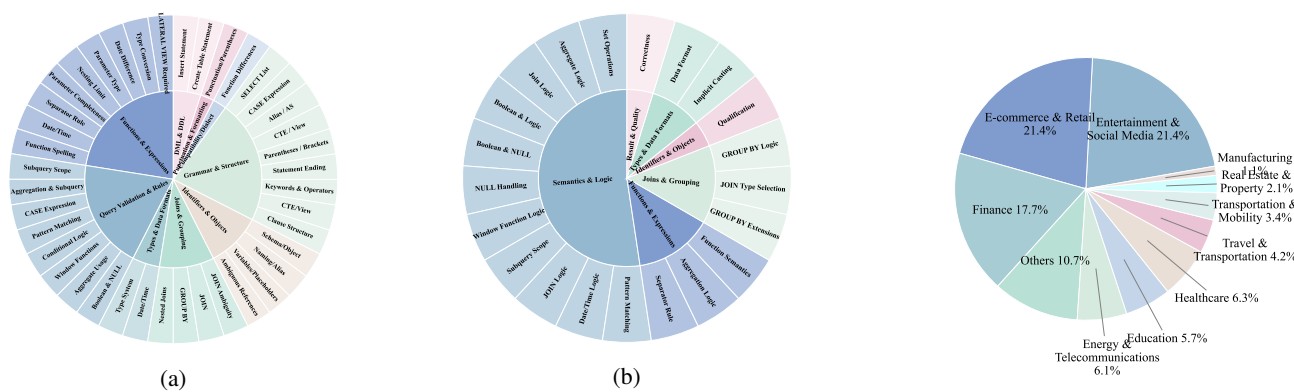

    (a)        (b)        (c)

*Figure 3.* Statistics of errors and domain distribution in Squirrel Benchmark. (a) Two-level error types in Squirrel-Syntax, highlighting the distribution of syntax errors. (b) Two-level error types in Squirrel-Semantic, showing the distribution of semantic errors. (c) Distribution of SQL code across different business domains.

**Hierarchical Error Taxonomy.** Figures 3(a) and (b) show the two-level error taxonomy for Squirrel-Syntax and Squirrel-Semantic. Detailed error type statistics are in Appendix I. Squirrel Benchmark covers a broad spectrum of common syntax and semantic errors, enabling fine-grained evaluation of model capabilities. Syntax errors include issues related to grammar, structure, and dialect, while semantic errors encompass type mismatches, aggregation errors, and logical inconsistencies. This hierarchical classification allows for detailed insight into model performance across error types, supporting a more rigorous assessment of debugging ability.

**Diversity of Task Scenarios.** As shown in Figure 3 (c), the domains in Squirrel Benchmark span finance, e-commerce, healthcare, and more than ten additional areas, ensuring that models are evaluated against a broad range of business logic and contextual dependencies. For example, a program from the financial domain may involve complex window func-

tions for time-series analysis, whereas an e-commerce program might require reasoning over multi-table joins across user and product schemas. This diversity tests a model's ability to generalize beyond simplistic syntactic patterns and demands domain-aware reasoning. Consequently, performance on Squirrel Benchmark provides a stronger indicator of a model's practicality and readiness for deployment in heterogeneous real-world environments.

## 5 Experiments

Detailed experimental settings are in Appendix D; this section highlights key results.

### 5.1 Main Results

**Existing LLMs are far from being experts on enterprise SQL debugging.** As shown in Table 2, we evaluate a diverse set of LLMs on Squirrel, including the Qwen,

*Table 2.* Evaluation results of LLMs on Squirrel-Syntax and Squirrel-Semantic. For each section, the best performance is highlighted in **bold**, and the second-best is underlined. EM, GM, and MB denote exact match score, graph match score, and modify-better score, respectively.

| Model | Size | Reasoning | MoE | Squirrel-Syntax | | | Squirrel-Semantic | | |
|---|---|---|---|---|---|---|---|---|---|
| | | | | EM | GM | MB | EM | GM | MB |
| *Open Source* | | | | | | | | | |
| Qwen-2.5-Instruct | 7B | | | 2.13 | 8.53 | 33.05 | 1.94 | 5.62 | 14.15 |
| Qwen-2.5-Coder | 7B | | | 3.20 | 8.96 | 37.53 | 4.84 | 7.75 | 18.99 |
| Qwen-2.5-Coder | 32B | | | 12.79 | 20.26 | 52.88 | **17.44** | **23.45** | **34.69** |
| Qwen-3-Instruct | 235B | | ✓ | 9.38 | 20.47 | 61.19 | 10.27 | 15.50 | 27.57 |
| Qwen-3-Coder-Instruct | 30B | | ✓ | 5.54 | 20.90 | 44.14 | 6.40 | 15.12 | 24.42 |
| Qwen-3-Coder-Instruct | 480B | | ✓ | 14.93 | 23.88 | 61.62 | 17.05 | 19.96 | 31.84 |
| QwQ | 32B | ✓ | | 8.76 | 20.51 | 41.45 | 10.47 | 15.31 | 20.16 |
| Seed-Coder-Instruct | 8B | | | 8.53 | 14.93 | 42.43 | 8.72 | 14.15 | 24.61 |
| OmniSQL | 32B | | | 0.21 | 6.40 | 50.75 | 0.39 | 6.40 | 21.17 |
| Deepseek-V3 | 685B | | ✓ | 17.91 | **30.28** | 60.34 | 11.24 | 21.32 | 33.27 |
| Deepseek-V3.1 | 685B | | ✓ | 17.91 | 30.49 | **63.61** | 12.02 | 14.73 | 32.47 |
| Deepseek-R1 | 671B | ✓ | ✓ | **18.34** | 21.98 | 58.64 | 15.89 | 22.09 | 30.14 |
| *Closed Source* | | | | | | | | | |
| Claude-4-Sonnet | — | ✓ | | **23.88** | **36.46** | **68.02** | **31.78** | **32.17** | **43.69** |
| GPT-4o-mini-2024-07-18 | — | | | 1.71 | 4.69 | 13.01 | 5.62 | 6.40 | 8.74 |
| GPT-4o-2024-11-20 | — | | | 2.14 | 4.69 | 13.79 | 2.91 | 4.84 | 6.86 |
| GPT-4.1 | — | | | 6.40 | 17.70 | 61.25 | 8.52 | 17.05 | 30.49 |
| GPT-5 | — | ✓ | | 13.43 | 18.55 | 66.52 | 16.28 | 16.47 | 29.90 |
| Gemini-2.5-Pro | — | ✓ | | 15.78 | 21.54 | 62.37 | 14.15 | 23.06 | 34.37 |
| Kimi-K2 | — | ✓ | ✓ | 14.07 | 27.72 | 61.83 | 15.70 | 20.93 | 31.84 |
| O1-preview | — | ✓ | | 8.32 | 21.11 | 46.27 | 8.14 | 11.43 | 14.43 |
| O3-mini | — | ✓ | | 3.84 | 19.83 | 63.54 | 10.47 | 28.68 | 40.78 |
| Doubao-Seed-1.6 | 230B | ✓ | ✓ | 19.19 | 30.92 | 64.39 | 16.09 | 20.93 | 32.82 |
| Doubao-Seed-1.6-flash | 230B | ✓ | ✓ | 1.50 | 3.63 | 9.62 | 1.55 | 3.11 | 6.42 |
| Doubao-Seed-1.6-thinking | 230B | ✓ | ✓ | 15.35 | 23.24 | 60.98 | 16.67 | 20.93 | 30.87 |
| *Comparison of different SFT method on Qwen-2.5-Coder* | | | | | | | | | |
| + SFT | | | | 26.44 | 30.70 | 48.40 | 14.34 | 15.70 | 18.02 |
| + diff-SFT | 7B | | | 22.17 | 22.81 | 34.33 | 7.95 | 9.30 | 12.60 |
| + DM-SFT | | | | **27.27** | **33.18** | **55.67** | **15.12** | **18.99** | **24.81** |

DeepSeek, Claude, GPT, Gemini, and Doubao families. Claude-4-Sonnet achieves the best performance, with a peak success rate of 36.46% GM score on Squirrel-Syntax and 32.17% GM score on Squirrel-Semantic. Other closed-source LLMs perform even worse, with most failing to exceed 20% GM. Among open-source models, DeepSeek-V3 achieves 30.28% on Squirrel-Syntax, and Qwen-2.5-Coder-32B attains 23.45% on Squirrel-Semantic, demonstrating competitive performance relative to closed-source systems.

**Code generation LLMs struggle with SQL debugging.** In previous studies, most code LLMs are heavily optimized for code generation, achieving strong performance on benchmarks such as SWE-Bench (Jimenez et al., 2024), BIRD (Li et al., 2024), and Spider (Yu et al., 2018). For example, OmniSQL (Li et al., 2025), a Text-to-SQL–specialized model,

achieves 87.6% on Spider and 64.5% on BIRD. However, its performance on Squirrel-Syntax and Squirrel-Semantic drops sharply to only 6.4% GM, underscoring the substantial gap between SQL generation and SQL debugging.

**Reasoning-oriented LLMs (RLMs) exhibit stronger refinement abilities.** Comparing RLMs with non-RLMs, we find that RLMs consistently perform better across both open-source and closed-source families. Notably, most RLMs achieve MB scores above 50%, indicating that while their predictions often move closer to the correct solution, they rarely solve the task in a single attempt.

**Squirrel-Semantic is more challenging than Squirrel-Syntax.** Across all evaluated models, performance on Squirrel-Semantic is consistently lower than on Squirrel-

Syntax. This is because Squirrel-Syntax provides explicit error messages, which help models localize faulty positions, whereas Squirrel-Semantic requires reasoning about deeper semantic inconsistencies without surface-level cues.

## 5.2 Can SFT Solve the SQL debugging?

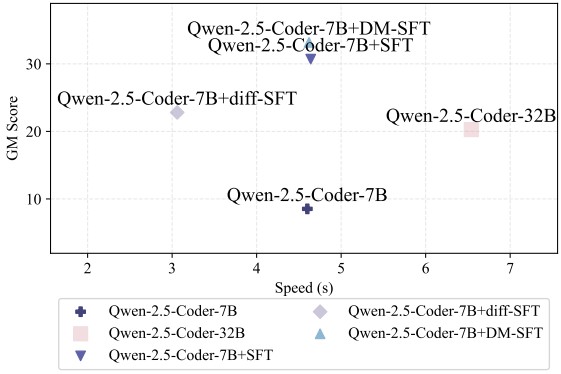

*Figure 4.* SFT baseline performance on Squirrel-Syntax. The horizontal axis represents the average inference speed, and the vertical axis shows the GM score.

As detailed in Appendix D.3.1 and Figure 7, we propose 3 representative SFT approaches as baselines: **(1) Vanilla SFT**, which directly fine-tunes the model on parallel SQL debugging pairs; **(2) DM-SFT** (Duan et al., 2025), which dynamically masking the loss for unchanged tokens in responses; **(3) Diff-SFT**, which frames SFT as a search-and-replace task, focusing only on the modified code segments. Results in Table 2 and Figure 4 shows:

**(1)** Targeted in-domain SFT significantly improves SQL debugging performance. Specifically, Qwen-2.5-Coder-7B + SFT substantially outperforms the base Qwen-2.5-Coder-7B, achieving a 33.17% gain in GM score on Squirrel-Syntax, and even surpasses Qwen-2.5-Coder-32B by 10.44%. **(2)** DM-SFT improves performance over vanilla SFT by masking the loss on non-diff tokens during training. This design forces the model to focus more on diff segments within pairs, thereby enhancing its effectiveness. **(3)** Diff-SFT predicts only the diff segments instead of generating the full code, offering a substantial inference speed advantage and reducing generation hallucination. On our benchmark, it requires only half the time of other methods, which is particularly beneficial for longer code snippets in enterprise applications. However, due to a mismatch between the search-and-replace task and the pretraining/SFT objectives of the base model, its GM score is slightly lower. Overall, these three SFT strategies provide strong baselines for future research on SQL debugging. More analysis is available in Appendix E.1.

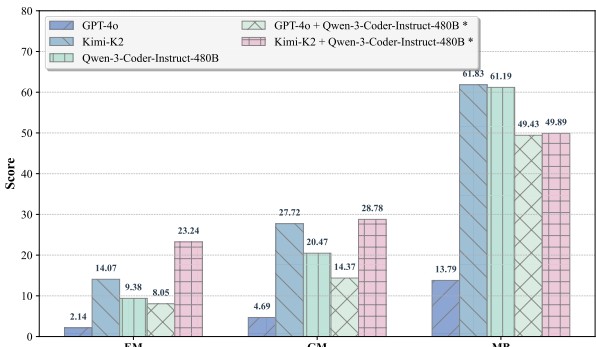

*Figure 5.* Agent performance on Squirrel-Syntax. '∗' denotes agent-based methods, while others are single-model baselines.

## 5.3 Can Agent Methods Solve the SQL debugging?

As shown in Appendix D.3.2 and Figure 8, we introduce an agentic baseline where a main agent plans SQL fixes and a sub-agent implements them, iterating with TQS[1] feedback until completion. Figure 5 shows that **agent-based systems can significantly boost performance, but results heavily depend on the main agent's capabilities.** For example, using Kimi-K2 as the main agent and Qwen3-Coder as the sub-agent increases EM accuracy by 65% compared to the Kimi-K2 single-model baseline. In contrast, when GPT-4o serves as the main agent—despite a 300%+ gain over its single-model performance—the combined system still underperforms the single Qwen3-Coder model. We also observe a decline in the MB score of agent-based systems, as multiple rounds of modification can gradually cause the model to deviate from the original SQL. These observations provide initial insights for future exploration of agentic methods in SQL debugging.

## 6 Related Work

The landscape of modern software engineering is increasingly shaped by the integration of LLMs to automate and augment developer tasks.

**Code Generation and Text-to-SQL Benchmarks.** Early text-to-code benchmarks, including HumanEval (Chen et al., 2021), SQL-Spider (Yu et al., 2018), and BIRD (Li et al., 2024), focus on simple and short code snippets (Zhuo et al., 2025; Jain et al., 2025; Liu et al., 2024). To address the gap with real-world applications, SWE-Bench (Jimenez et al., 2024) evaluates models on complete software issues, which require a comprehensive understanding of codebases. Similarly, Spider2.0 (Lei et al., 2025) extends Text-to-SQL evaluation to enterprise contexts. BIRD-Critic (Li et al., 2024) introduces SQL debugging, but it only handles short,

---

[1]The TQS tool is introduced in Appendix D.1.3.

simplified StackOverflow queries that lack enterprise-level complexity. Most of these benchmarks rely on manually curated datasets, which are costly and prone to data leakage (Zhou et al., 2026). In this work, we introduce the first enterprise-level SQL debugging benchmark, which is automatically constructed via reverse engineering.

**LLMs for Automated Software Engineering.** Recent work applies LLMs to automated software engineering through three primary paradigms: (1) **Single-model** approaches, which attempt to produce patches directly from a description and buggy code, often using few-shot prompting or SFT (Huang et al., 2024; Yasunaga & Liang, 2021; Allamanis et al., 2021). These single-model methods are bottlenecked by the need to build large-scale SFT datasets (Pan et al., 2024; Li et al., 2026; Ma et al., 2024; Yang et al., 2026; Pham et al., 2025). (2) **Multi-stage Workflows**, which guide models through defect localization, patch generation, and validation (Xia et al., 2025; Zhang et al., 2024; Gong et al., 2026). (3) **Agent-based Methods**, which leverage analysis, execution traces, or test feedback for iterative refinement (Yang et al., 2024; Wang et al., 2025; Bouzenia et al., 2025; Chen et al., 2024b). In this work, we provide both SFT-based Single-model solutions and Agent-based methods, offering the community a comprehensive understanding of SQL debugging tasks.

## 7 Conclusion

We introduce Squirrel Benchmark, the first benchmark for enterprise-level SQL debugging. With its automated construction workflow and execution-free evaluation, Squirrel Benchmark enables scalable and reliable assessment of LLMs. Despite recent advances in LLM reasoning, our evaluation of nearly 30 models shows that real-world enterprise SQL debugging remains a significant challenge. To encourage further progress, we highlight four promising directions, including three SFT-based strategies and one agent-driven approach. Importantly, Squirrel Benchmark correlates strongly with practical debugging performance, making it a reliable reference for both academic research and industrial deployment.

## Impact Statement

This paper presents work whose goal is to advance the field of Machine Learning. There are many potential societal consequences of our work, none of which we feel must be specifically highlighted here.

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

# Appendix

## A    Use of LLMs

We disclose the following uses of LLMs in this work:

1. **Data Construction.** LLMs were used during benchmark and dataset construction. Details of the data generation process are provided in Section 3, and the corresponding prompts are included in Appendix H.

2. **Benchmark Evaluation.** LLMs were evaluated on the benchmark introduced in this paper. The evaluated models are listed in Appendix D.2.

3. **Manuscript Preparation.** LLMs were used exclusively for language editing, proofreading, and stylistic refinement of the manuscript.

4. **Programming Assistance.** LLMs were used to assist with software development. All generated code was manually reviewed and validated by the authors.

All research ideas, methodological designs, experimental decisions, analyses, and scientific conclusions presented in this work were conceived and developed solely by the authors. No LLM contributed to the intellectual content or scientific claims of this research.

## B    Background of ETL SQL debugging.

Our benchmark targets industrial Extract–Transform–Load (ETL) workloads, which differ substantially from traditional Text-to-SQL or analytics-oriented SQL generation. We summarize the key distinctions below.

**Task objectives and nature.** ETL is primarily a data engineering task focused on preparing, transforming, and integrating raw data into a clean and consistent warehouse for downstream consumption. The goal is reliable data production rather than interactive analysis. In contrast, Data Analysis / Text-to-SQL tasks aim to explore existing datasets and answer analytical questions. These tasks resemble the work of data analysts: flexible, insight-driven, and focused on extracting knowledge from already-curated data rather than producing new datasets.

**Data scale and operations.** ETL pipelines operate on full-scale production data. For example, computing daily active users may require scanning and joining hundreds of millions of raw event records. The dominant SQL operations involve `INSERT`, `UPDATE`, `DELETE`, and `MERGE`, reflecting an emphasis on data movement, reshaping, and materialization. By contrast, Data Analysis / Text-to-SQL workloads typically query curated warehouse tables using complex `SELECT` statements that return relatively small result sets—reports, leaderboards, or summary statistics. These tasks focus on the correctness of the query output rather than on large-scale data transformation.

## C    Source of SQL Bug Taxonomy.

To construct the SQL bug taxonomy, we analyze a corpus of production logs and select 268 representative samples for detailed manual inspection. Three domain experts—each with more than three years of professional experience in SQL analysis and data quality verification—annotate these logs over a two-week period. The error categories identified in these annotations form the basis of the final SQL Bug Taxonomy.

## D    Experimental Settings

### D.1    Evaluation

#### D.1.1    CHALLENGES IN EXECUTION-BASED EVALUATION

Evaluating enterprise SQL generation and debugging systems presents several unique challenges. *First*, conventional execution-based accuracy—where correctness is determined by comparing a program's output to a reference—is often impractical in production environments. This is due to two primary constraints: (1) **Security and Privacy**: Production

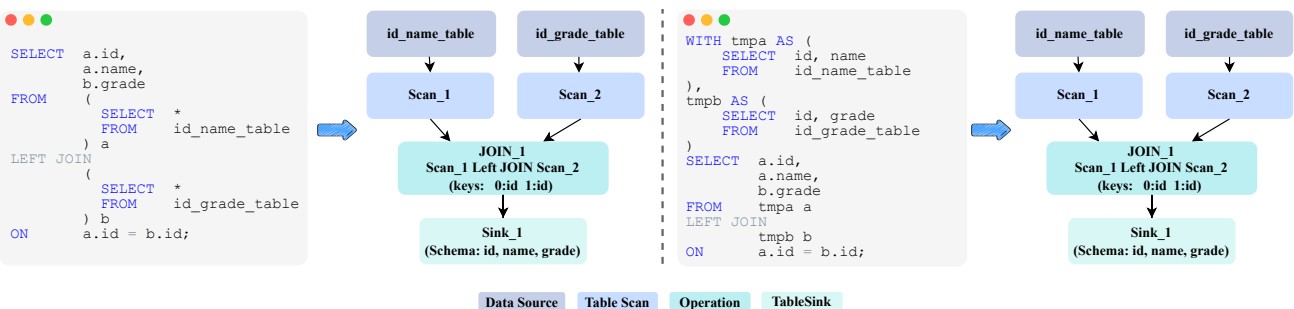

*Figure 6.* Illustration of Graph Match Score. Although the left and right SQL snippets differ syntactically, their optimized abstract syntax trees are structurally identical. Graph matching evaluates semantic equivalence through tree isomorphism.

databases typically contain proprietary or sensitive data, making arbitrary code execution infeasible; (2) **Efficiency**: Executing complex SQL scripts on large-scale production datasets is computationally expensive and time-consuming. *Second*, correctness is not binary. Unlike in standardized benchmarks, real-world SQL debugging admits multiple valid solutions. A repair can be correct through various syntactic paths or logical approaches. *Third*, string-based metrics are a poor proxy for quality. Comparing predicted SQL to a reference string ignores functional equivalence. Therefore, a robust evaluation framework must balance efficiency and accuracy while reliably reflecting SQL quality in real-world problem-solving.

### D.1.2 EVALUATION METRICS

To address challenges in SQL debugging evaluation, we introduce an execution-free evaluation methodology based on three complementary metrics.

**Exact Match Score (EM).** This metric provides a strict, reproducible measure of syntactic correctness by comparing the predicted SQL string directly against the reference:

$$\text{EM} = \frac{1}{N} \sum_{i=1}^{N} \mathbf{1}[\hat{q}_i = q_i] \tag{5}$$

where $\hat{q}_i$ is the predicted SQL, $q_i$ is the reference SQL, and $\mathbf{1}[\cdot]$ is the indicator function. While stringent, EM serves as a clear lower bound on model performance.

**Graph Match Score (GM).** To assess the semantic equivalence of SQL queries, we represent each query as an optimized abstract syntax tree, illustrated in Figure 6. Each node corresponds to a logical relational operator (e.g., *Join, Project, Filter*), and the hierarchical structure encodes operator dependencies and execution order. We use Apache Calcite (Begoli et al., 2018) to compile SQL queries into its canonical intermediate representation. Calcite applies a suite of logical rewrites—such as operator reordering and clause simplification—to produce normalized logical plans that are robust to superficial syntactic differences. This intermediate form also encodes both control and data dependencies as edges, yielding graph structures that capture deeper aspects of query semantics. These enriched graphs enable more faithful comparison and interpretation of SQL behavior. The GM score is computed by performing graph isomorphism over the normalized representations. This allows our method to detect semantic equivalence even when queries differ substantially in surface form:

$$\text{GM} = \frac{1}{N} \sum_{i=1}^{N} \mathbf{1}[\text{Graph}(\hat{q}_i) \cong \text{Graph}(q_i)] \tag{6}$$

where $\cong$ denotes graph isomorphism. This approach recognizes semantically equivalent codes that may differ syntactically.

**Modify Better Score (MB).** For iterative debugging scenarios, absolute correctness is insufficient; we must measure progressive improvement. The MB metric evaluates whether a prediction moves closer to the correct solution by comparing AST edit distances:

$$\text{MB} = \frac{1}{N} \sum_{i=1}^{N} \mathbf{1}[d(\hat{q}_i, q_i) < d(b_i, q_i)] \tag{7}$$

where $d(\cdot, \cdot)$ denotes normalized AST edit distance, $\hat{q}_i$ is the predicted repair, $q_i$ is the reference SQL, and $b_i$ is the original buggy query. This metric specifically assesses a model's capacity for incremental repair in debugging workflows.

Together, these metrics provide a comprehensive evaluation framework that balances efficiency, reproducibility, and semantic understanding while avoiding the practical limitations of execution-based assessment.

### D.1.3 EXECUTION-BASED VALIDATION

The earlier discussion on the impracticality of execution accuracy may appear to conflict with the execution checks referenced in this paper. To clarify, all execution-based validation in our work is strictly non-executive—that is, we do not run SQL scripts against a live engine. Instead, we rely on TQS, an enterprise-grade SQL quality validation tool built on Apache Calcite, which performs comprehensive static analysis, including **syntax and semantic checks** to ensure scripts are syntactically valid and logically well-formed, as well as **schema and column validation** to confirm that all referenced tables and fields exist in the physical schema. This static-analysis approach provides rigorous error detection during development while avoiding the practical limitations associated with true execution-based evaluation.

### D.1.4 RELIABILITY OF GRAPH MATCH SCORE

Graph Match (GM) is a relatively strict metric that can occasionally classify correct outputs as incorrect. To assess its reliability, we conducted a human evaluation on 300 samples, which shows that GM agrees with expert judgments in over 68% of cases. While such omissions can occur, they tend to affect all models fairly. To further validate GM, we analyzed manual annotations on 100 samples for each of three different models. The results indicate that the model ranking based on GM scores is consistent with the ranking derived from human evaluation. This consistency provides evidence that GM scores are a reasonably reliable metric for comparing model performance.

### D.2 LLMs

This study ensures a robust evaluation by leveraging a diverse set of LLMs, encompassing both open-source and proprietary architectures to cover a broad range of capabilities.

**Open-Source Models.** DeepSeek-R1-0528 (DeepSeek-AI, 2025b), DeepSeek-V3-0324 (DeepSeek-AI, 2025a), DeepSeek-V3.1, Qwen-2.5-Coder (Hui et al., 2024), Qwen-3-235B-A22B-Instruct-2507 (Yang et al., 2025), Qwen-3-Coder-480B-A35B-Instruct (Qwen, 2025), QwQ-32B, Seed-Coder-8B (Seed et al., 2025), and OmniSQL-32B (Li et al., 2025).

**Closed-Source Models.** Claude-Sonnet-4 (Anthropic, 2025), GPT-4o-mini-2024-07-18, GPT-4o-2024-11-20 (OpenAI, 2024), GPT-4.1 (OpenAI, 2025), o3-mini (OpenAI, 2025), o1-Preview, GPT-5, Gemini 2.5 Pro (Gemini, 2025), Kimi-K2 (Kimi-Team, 2025), and Doubao-Seed-1.6, Doubao-Seed-1.6-Flash, and Doubao-Seed-1.6-Thinking (Seed, 2025).

### D.3 Baselines

### D.3.1 SFT BASELINES

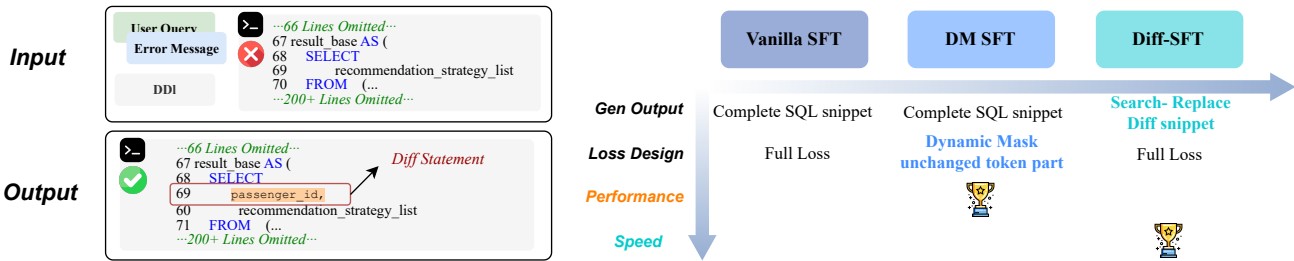

*Figure 7.* Illustration of three distinct supervised fine-tuning (SFT) methods.

We propose three distinct supervised fine-tuning (SFT) methods as baselines.

**Vanilla SFT.** This is the standard sequence-to-sequence fine-tuning approach. The model takes as input the error message, the DDL, and the issue SQL, and is trained to generate the complete, corrected reference SQL. While simple, this method establishes a fundamental baseline for performance.

**DM-SFT (Dynamic-Masked SFT).** In enterprise SQL debugging, the differences between an issue SQL and its reference SQL are often minimal within lengthy code snippets. Consequently, Vanilla SFT models can rapidly reduce loss by learning to copy the large, unchanged portions of the input, potentially failing to focus on the critical, erroneous segments. To mitigate this, we adopt Dynamic-Masked SFT (DM-SFT) (Duan et al., 2025), which randomly masks the loss calculation for 50% of the tokens that are identical between the input and output. By increasing the loss contribution of the changed tokens, this method encourages the model to prioritize learning the necessary edits.

**Diff-SFT.** Generating the complete SQL code significantly increases inference overhead. We propose an alternative method that outputs only a "diff" snippet, framing the task as a search-and-replace operation. The model's objective is to identify the erroneous code segment in the input and generate the corresponding corrected snippet.

### D.3.2 AGENT BASELINES

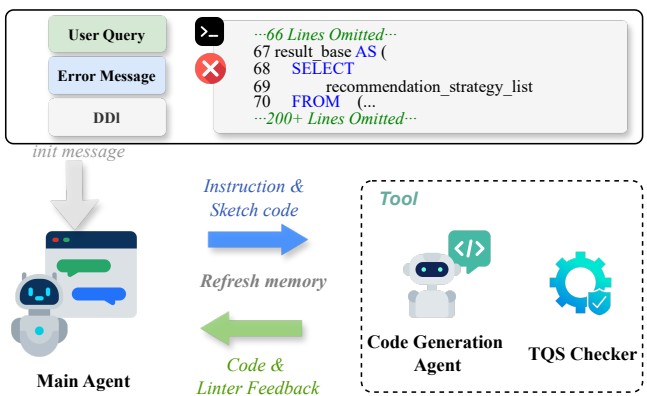

*Figure 8.* Overview of the agentic method, which consists of a main agent, a code-generation sub-agent, and a TQS checker tool. The *main agent* observes the error message and the issued SQL, analyzes the cause of the failure and the required modification, and generates a code-editing instruction for the code generation sub-agent. The *code generation sub-agent* applies the instruction to modify the code; the updated code is automatically passed through a TQS checker to detect errors, and the resulting code and lint feedback are used to update the main agent's memory. This iterative loop continues until the main agent determines that all necessary modifications have been completed.

As illustrated in Figure 8, we design an **agentic framework** that coordinates multiple specialized components to iteratively refine generated SQL queries and resolve execution failures. The framework consists of three modules: a main agent, a code-generation sub-agent, and a TQS tool.

The main agent serves as the central controller. It receives the error message and the proposed SQL code for the issue from the previous iteration. Based on these inputs, it analyzes the underlying cause of failure, identifies the modifications required to fix the issue, and produces a structured instruction describing the intended code change. This instruction is sent to the code-generation sub-agent. The code-generation sub-agent performs the actual code editing. It interprets the modification instruction and updates the SQL sketch code accordingly. Once the revision is complete, the generated code is automatically processed by a TQS checker, which detects syntax errors, style violations, and structural inconsistencies. The resulting code and lint feedback are then incorporated into the main agent's memory. This interaction forms an iterative correction loop. The main agent continuously observes the updated code and diagnostic feedback, issuing refined modification instructions until it concludes that the SQL query is correct and no further edits are required.

### D.4 Dataset

Our training dataset consists of three components:

- **Reverse-engineered data**: We manually injected bugs into correct SQL queries collected from production logs, yielding a total of $2,015$ samples.

- **Log-mined data**: We extracted erroneous SQL queries and their associated error messages from online execution logs. For each instance, the reference SQL was manually written and validated by domain experts, resulting in $1,971$ samples.

- **Synthetic data**: We generated additional samples from the BIRD (Li et al., 2024) and Spider (Yu et al., 2018) Text-to-SQL datasets to expand the SFT data, producing $1,054$ samples.

### D.5 Hyperparameters

**Fine-tuning.** For self-supervised fine-tuning, models are trained for $5$ epoch with a learning rate of $1e-5$ and a per device batch size of $64$. We employed the AdamW optimizer and a cosine learning rate scheduler with a warm-up phase corresponding to $3\%$ of the total training steps.

**Evaluation.** Following Yang et al. (2024); Chen et al. (2024a), we use a temperature of $0.0$ for deterministic action decoding and input truncation to manage context length.

### D.6 Experimental Environments

Our code primarily relies on Python 3.12 and PyTorch 2.7.0. Models are self-supervised fine-tuned with `LLaMA-Factory` (Zheng et al., 2024) [2], and inference is performed with `vLLM` (Kwon et al., 2023).

## E Additional Experimental Results

### E.1 Additional Analysis of SFT Performance on Squirrel Benchmark

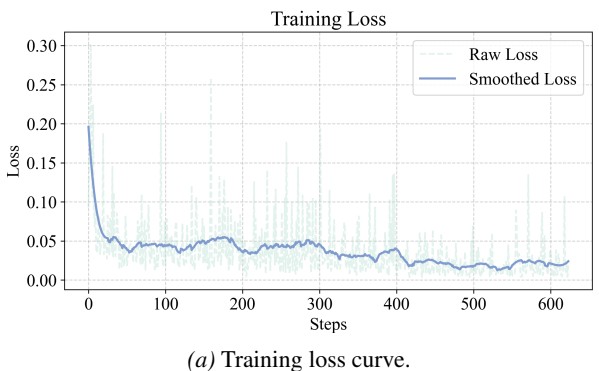

*(a)* Training loss curve.

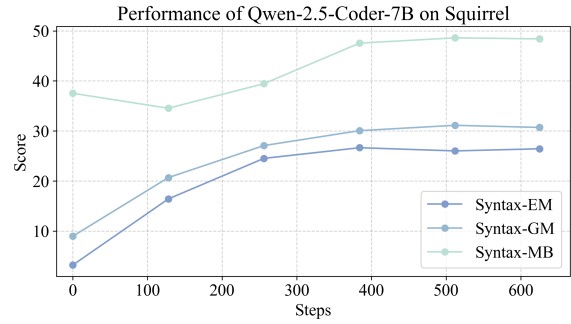

*(b)* Performance at different training steps.

*Figure 9.* Analysis of Qwen-2.5-Coder-7B Vanilla SFT on Squirrel Benchmark, showing corresponding training loss and step-wise performance.

**Rapid loss decay in SQL debugging fine-tuning.** Figure 9a illustrates that the training loss quickly drops below 0.05 within a few steps, approaching zero. This behavior arises because the constructed SQL debugging parallel data contain inputs with error messages and issue SQL statements, and outputs with the corrected SQL. In most cases, only a small portion of tokens differ between the input and output. Consequently, the model primarily copies tokens from the input, leading to extremely low training loss. When the majority of output tokens carry minimal information, the model tends to ignore the truly informative segments that require correction.

**Performance improves with increased training steps.** Figure 9b shows that as training progresses, model performance steadily improves, particularly during the early steps. Beyond approximately 400 training steps, the gains become marginal, indicating diminishing returns. This trend suggests that while additional in-domain training helps, the benefit of further fine-tuning eventually saturates.

## F Limitations

This work introduces a benchmark for enterprise SQL debugging, providing a foundation for future research in software engineering. However, several limitations remain.

---

[2]https://github.com/hiyouga/LLaMA-Factory.git

**First, the synthetic nature of the benchmark.** Although the dataset is automatically generated by LLMs, we manually inspected and cross-validated cases that all models failed (detailed in Section 3.4). Nevertheless, undetected artifacts may still exist. Developing more robust automated validation methods is an important direction for future work.

**Second, constraints of the evaluation framework.** Our rule-based, execution-free evaluation combines exact match, graph match, and edit-direction criteria (detailed in Appendix D.1.2). While effective for debugging scenarios where minimal and precise fixes are expected, this approach is inherently limited by its reliance on reference solutions. For more semantic tasks in which solutions may vary widely, a more flexible and semantics-aware evaluation methodology is needed. We identify this as an area for improvement.

**Third, the limited SQL dialect coverage.** Squirrel Benchmark is currently built on Hive/Spark SQL, one of the most widely adopted dialects in large-scale enterprise data infrastructures. Although broader dialect coverage would be valuable, our construction methodology is fundamentally dialect-agnostic, allowing datasets to be synthesized for other SQL dialects. Future iterations will explore additional dialects to expand the benchmark's coverage.

# G Case Study

| | |
|---|---|
| **Error Message:** | **org.apache.calcite.sql.parser.SqlParseException: Encountered "AS" at line 14, column 54. Was expecting one of: ")" ..."MULTISET" ... "ARRAY" ...** |

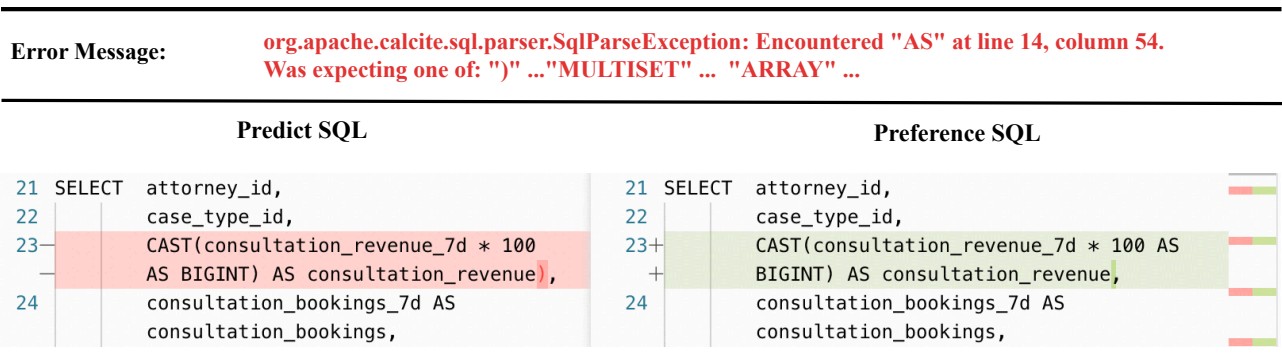

*Figure 10.* **Model Hallucination:** After modifying the code according to the error message, the model also inserted an extra ")" in similar fragments, which caused the fix to fail.

| **Error Type:** | Level 1 Error Type | Level 2 Error Type | Level 3 Error Type |
|---|---|---|---|
| | Query Validation & Rules | Subquery Scope | Outer query references alias not visible in subquery |

| **Error Message:** | **org.apache.calcite.runtime.CalciteContextException:: at line 233:37: Table 'b' not found** |
|---|---|

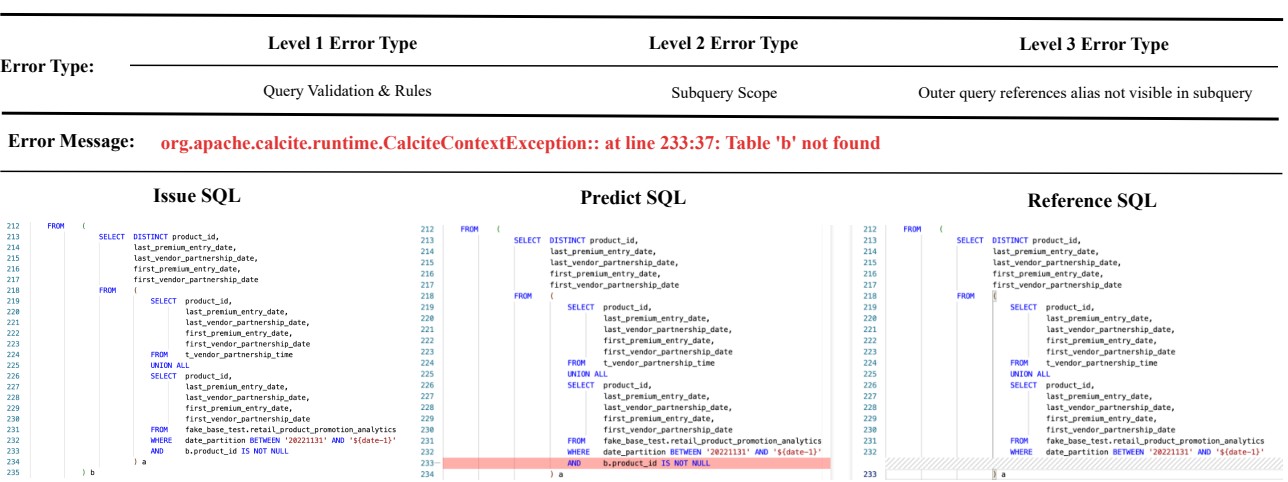

*Figure 11.* **Long Context Reasoning Limitation:** The error code uses a non-existent table b (which is usually an alias for a longer table name in SQL), but the model fails to detect this error during the repair process.

# H Prompts Template

The detailed prompts are described below.

## H.1 Enterprise-level SQL Scripts Generation Prompts

> **Prompt for Scenario Creation**
>
> ## Instruction
>
> You are a professional SQL ETL and schema generation expert. Your task is to transfer a database schema from a source domain to a target domain, preserving structural complexity and table relationships, but fully adapting table names, field names, and semantics to the target domain.
>
> ## Steps
>
> 1. Analyze Source DDL:
> - Examine the number of tables, fields, data types, relationships, and naming patterns.
> - Treat this as a structural seed for generating an equivalent schema.

2. Generate Target Schema:
- Create a logically equivalent schema under the target domain.
- Rules:
- Use the database `fake_base_test`.
- Format: CREATE TABLE IF NOT EXISTS `fake_base_test.table_name ( ... )`;
- Avoid SQL reserved keywords as column names.
- Reflect business meaning in the target domain.
- Optionally add auxiliary fields to maintain equivalent complexity.
- All names, comments, and logic must be consistent with the target domain and unrelated to the source domain.

3. Validation:
- Ensure DDL syntax is correct.
- Ensure schema and scenario are fully adapted to the target domain, with no remnants from the source.

## Notes
- Do not reuse proprietary identifiers or field names from the source domain.
- Only use the user-provided target domain.
- Preserve the structural pattern, complexity, and relationships of the source schema.

## Input Data

Source DDL: `DDL`
Target Domain: `SCENARIO`

## Output Format(JSON)

```
{
    "mock scenario": "Scenario description",
    "mock ddl": "Corresponding CREATE TABLE statements"
}
```

---

## Prompt for Generating Enterprise-level SQL

## Instruction

You are a professional SQL ETL code generation expert. Using the provided source SQL as a reference, and given the target domain scenario and its corresponding DDL, generate an SQL ETL script for the target domain that preserves the logical structure and complexity of the source code while adapting it fully to the target domain.

## Requirement

1. Logical structure equivalence:
- Analyze the ETL workflow, table relationships, and processing steps in the source SQL code.
- Preserve the overall structure, complexity, and transformation logic, but replace all table names, field names, and data types to match the target domain.

2. Strictly match the target DDL:
- All SQL must be fully based on the provided target DDL.
- Table names and field names must match the target DDL exactly.
- Do not retain any original business terms, identifiers, or domain concepts from the source code.

3. Output requirements:
- The code must be executable, and SQL syntax must be correct.
- Maintain a clear hierarchy and readability (include appropriate comments).
- Naming should reflect the target business domain, ensuring a one-to-one correspondence between SQL and the target DDL.

## Input Data

Source SQL: `SQL`
Target Domain Scenario: `SCENARIO`
Target DDL: `DDL`

```
## Output Format

{
    'mock code': 'Generated target domain SQL ETL code'
}
```

## H.2  Issue SQL Construction Prompts

**Prompt for Error Type Selection**

## Role:
You are an expert SQL engineer specializing in designing realistic SQL bugs for testing and debugging scenarios.

## Task:
Given a correct SQL query, your job is to:
Select the top {TOP_K} appropriate error type from the provided taxonomy.

##Key Guidelines:
- Minimal Change: Only introduce the chosen bug. Do not alter the original query's structure or intent more than necessary.
- Realism: The bug should reflect mistakes that real developers are likely to make.

##Input:
1. Correct SQL: {SQL}
2. DDL (optional): {DDL}
3. Original Intent: {CODE INTENTION}
4. Error Type Taxonomy: {SEMANTIC ERROR TYPES}

##Output Requirements:
Your output must include:
- The selected error type(s) at Level 1–3 granularity.

##Output Format:
```
{
    candidate_errors:
        {
            "level1_error_type": Level 1 error type,
            "level2_error_type": Level 2 error type,
            "level3_error_type": Level 3 error type
        },
        {
            "level1_error_type": Level 1 error type,
            "level2_error_type": Level 2 error type,
            "level3_error_type": Level 3 error type
        },
    ]
}
```

**Prompt for Squirrel-Syntax Issue SQL Construction**

## Role:
You are an expert SQL engineer specializing in designing realistic SQL bugs for testing and debugging scenarios.

## Task:
Given a correct SQL query, your task is to introduce an error into the correct query with the smallest possible change.

## Key Guidelines:
- Minimal Change: Only introduce the chosen bug. Do not alter the original query's structure or intent more than necessary.
- Realism: The bug should reflect mistakes that real developers are likely to make.

## Input:
1. Correct SQL: `{SQL}`

2. DDL (optional): `{DDL}`

3. Original Intent: `{CODE INTENTION}`

4. Error Type Taxonomy: `{SEMANTIC ERROR TYPES}`

## Output Requirements:
Your output must include:
- The selected error type(s) at Level 1–3 granularity.
- The modified SQL query with the injected bug.

## Output Format:

```
{
    "level1_error_type": Level 1 error type,
    "level2_error_type": Level 2 error type,
    "level3_error_type": Level 3 error type,
    "issue_sql": SQL query with the injected bug
}
```

---

### Prompt for Squirrel-Semantic Issue SQL Construction

## Role:
You are an expert SQL engineer specializing in designing realistic SQL bugs for testing and debugging scenarios.

## Task:
Given a correct SQL query, your job is to:
1. Introduce the error into the SQL query with the smallest possible change.
2. Write a realistic user-style issue report describing how the bug causes the query to behave incorrectly, and the user's real intention.

## Key Guidelines:
- Minimal Change: Only introduce the chosen bug. Do not alter the original query's structure or intent more than necessary.
- Realism: The bug should reflect mistakes that real developers are likely to make.

## Input:
1. Correct SQL: `{SQL}`

2. DDL (optional): `{DDL}`

3. Original Intent: `{CODE INTENTION}`

4. Error Type Taxonomy: `{SEMANTIC ERROR TYPES}`

## Output Requirements:
Your output must include:
- The selected error type(s) at Level 1–3 granularity.
- The modified SQL query with the injected bug.
- A natural-language user bug report describing the mismatch between expected and actual results (without exposing SQL code, since the user does not know the root cause).

## Output Format:

```
{
    "level1_error_type": Level 1 error type,
    "level2_error_type": Level 2 error type,
    "level3_error_type": Level 3 error type,
    "user_query": Bug report written in natural language.
```

```
      Describe the expected vs. actual outcome clearly.
      "issue_sql": SQL query with the injected bug
}
```

## H.3 Benchmark Evaluation Prompt

---

**Prompt for Squirrel-Syntax Generation**

You are an SQL assistant.

## Task

Based on the error messages and table schema, your task is to fix the issue in the SQL and write the correct SQL.
Remember that you can not change any existing comments and SQL code without errors.

## Input Data
The issue SQL: `BUG SQL`
Related tables schema: `DDL`
Error Messages: `ERROR MESSAGE`

## Output (JSON):

```
{
    'predict_sql': The fixed SQL.
}
```

---

**Prompt for Squirrel-Semantic Generation**

You are an SQL assistant.

## Task

Based on the user query and input table schema, please fix the bugs in the Issue SQL and write the corresponding correct SQL code.
Remember that you can not change any existing comments and SQL code without errors.

## Input Data
User Query:`USER QUERY`
Related tables schema: `DDL`
Error Messages: `ERROR MESSAGE`

## Output (JSON):

```
{
    'predict_sql': The fixed SQL.
}
```

---

**Prompt for diff Generation**

```
<background_info>
\texttt{DDL_PLACEHOLDER}
</background_info>

```code
SQL_CODE_PLACEHOLDER
```

---

```
```

<error_msg>
ERROR_MESSAGE_PLACEHOLDER
</error_msg>

```last_edit
<<<<<<< SEARCH
LAST_EDIT_BEFORE_PLACEHOLDER
=======
LAST_EDIT_AFTER_PLACEHOLDER
>>>>>>> REPLACE
```
```

## H.4  Agent Prompt

---

**Prompt for Main Agent**

You are a SQL expert. Please review the SQL code (with the table DDL) and the error message reported. Your task is to analyze the error and provide fixing edit instructions.

**Input:**
- Tables DDL
DDL_PLACEHOLDER
- Hive SQL Code:
```sql  SQL_CODE_PLACEHOLDER```
- Error Message:
ERROR_MESSAGE_PLACEHOLDER

**Output Requirements:**
You must strictly follow this XML format in your response:

<analysis>
Examine the error message and identify the root cause. Explain what is wrong with the current code and why the error occurred.
</analysis>

<instructions>
Provide clear, step-by-step instructions on how to fix the code. Explain what changes need to be made and where they should be applied.
</instructions>

<sketch_sql>
Provide the edit sketch using the special comment `...` to represent unchanged code between edited lines. Specify each edit in sequence, minimizing unchanged SQL code while making it clear what the edit is and where it should be applied.
</sketch_sql>

Ensure your instructions(in Chinese) and sketch are clear enough that another model can apply them correctly without accidentally deleting or modifying unintended parts of the code.

---

# I  SQL Bug Taxonomy

## I.1  Bug Distribution of Squirrel-Semantic

*Table 3.* Error type distribution in Squirrel-Semantic

| Level 1 Error Type | Level 2 Error Type | Level 3 Error Type | Count |
|---|---|---|---|
| Semantics & Logic | Aggregate Logic | Using COUNT(column) instead of COUNT(*) and misunderstanding NULL exclusion | 43 |
| | | Using SUM()/AVG() on a column with NULLs without COALESCE | 27 |
| | Join Logic | JOIN condition placed in WHERE clause (accidental CROSS JOIN) | 41 |
| | | Failing to handle NULLs in JOIN keys (causing rows to disappear) | 13 |
| | | Missing condition causing Cartesian product | 2 |
| | Boolean & Logic | Three-valued logic error: NOT (a = b) not equivalent to a != b when NULLs present | 14 |
| | | Improper Boolean usage (e.g., WHERE col = TRUE) | 9 |
| | NULL Handling | NULL compared with = (should use IS NULL) | 33 |
| | | Confusion between IS NULL and =NULL | 2 |
| | Window Function Logic | Using RANK() instead of ROW_NUMBER() or DENSE_RANK() leading to duplicates/skips | 31 |
| | | Incorrect partitioning/ordering in window function leading to wrong row assignment | 4 |
| | Subquery Scope | Misplaced LIMIT inside subquery affecting outer results | 3 |
| | | Correlated subquery missing correlation condition | 2 |
| | JOIN Logic | Missing condition causing Cartesian product | 12 |
| | | Wrong join key used inside nested subquery | 2 |
| | Set Operations | UNION vs. UNION ALL misuse (unintended deduplication) | 55 |
| | Date/Time Logic | Confusion between DATE, TIMESTAMP, and INTERVAL types | 23 |
| | Pattern Matching | Incorrect LIKE usage | 2 |
| Functions & Expressions | Separator Rule | collect_set/concat_ws separator uses semicolon | 54 |
| | Function Semantics | Misunderstanding the empty handling of aggregate functions | 1 |
| Joins & Grouping | GROUP BY Extensions | Misuse of ROLLUP / CUBE | 14 |
| | | Rollup/Cube/Grouping Sets producing unexpected super-aggregate rows | 3 |
| | GROUP BY Logic | Grouping by a functionally dependent column unnecessarily | 17 |
| | | Rollup/Cube/Grouping Sets producing unexpected super-aggregate rows | 4 |
| | JOIN Type Selection | Using INNER JOIN when LEFT JOIN is needed (loss of data) | 64 |
| Result & Quality | Correctness | Duplicate rows due to many-to-many join not being accounted for | 1 |
| | | Incorrect output data | 1 |
| Types & Data Formats | Implicit Casting | Implicit cast changing semantics (e.g., string to number) | 15 |
| | Data Format | Misused format placeholder | 1 |
| Identifiers & Objects | Qualification | Qualifying a column with the wrong table alias in a complex join | 22 |

## I.2  Bug Distribution of Squirrel-Syntax

# J  Examples

*Table 4.* Error type distribution in Squirrel-Syntax

| Level 1 Error Type | Level 2 Error Type | Level 3 Error Type | Count |
|---|---|---|---|
| Functions & Expressions | Parameter Completeness | Missing parameter for explode | 15 |
| | | Incorrect explode parameter | 9 |
| | | explode(map) requires two aliases | 2 |
| | | date_add missing parameter (also typo data_add) | 2 |
| | | array_contains wrong argument type | 1 |
| | Parameter Type | get_json_object wrong argument type | 4 |
| | | array_contains wrong argument type | 3 |
| | | from_json wrong argument type | 1 |
| | LATERAL VIEW Required | Missing LATERAL VIEW | 94 |
| | | Missing alias for LATERAL VIEW function output | 1 |
| | Date Difference | datadiff argument/typo error | 5 |
| | Type Conversion | Multiple AS in CAST | 15 |
| | Nesting Limit | Aggregate expressions cannot be nested | 2 |
| | Separator Rule | collect_set/concat_ws separator uses semicolon | 11 |
| | Date/Time | to_unix_timestap typo | 1 |
| | Function Spelling | concat_ws typo | 1 |
| Query Validation & Rules | CASE Expression | Missing END or THRN in CASE WHEN | 72 |
| | | Multiple END in CASE WHEN | 4 |
| | Conditional Logic | Missing argument in IN | 7 |
| | | IN subquery returns multiple columns | 1 |
| | Window Functions | Window function misused with GROUP BY | 3 |
| | | Window function used inside WHERE/HAVING | 3 |
| | | Window function frame clause misuse (e.g., ROWS BETWEEN error) | 1 |
| | Subquery Scope | Outer query references alias not visible in subquery | 2 |
| | Aggregation & Subquery | SELECT list contains non-aggregated column not in GROUP BY | 62 |
| | Pattern Matching | Incorrect LIKE usage | 2 |
| | Aggregate Usage | Aggregate function in SELECT without GROUP BY | 1 |
| | Boolean & NULL | NULL compared with = (should use IS NULL) | 2 |
| Grammar & Structure | Clause Structure | Incorrect clause ordering - JOIN after WHERE | 7 |
| | | Invalid SELECT clause syntax with subquery | 6 |
| | | Missing SELECT before FROM clause | 5 |
| | | Multiple WHERE | 4 |
| | | Missing partition conditions in WHERE clause | 3 |
| | | Missing logical connector in WHERE | 26 |
| | | Non-query expression in illegal context | 3 |
| | | Missing FROM clause | 2 |
| | | Column count mismatch in UNION | 1 |
| | CTE/View | WITH AS not first | 26 |
| | | Unnecessary WITH AS | 13 |
| | | Trailing comma after last view | 23 |
| | Keywords & Operators | Keyword spelling error | 3 |
| | | Space in != | 2 |
| | | Missing IN keyword | 2 |
| | Statement Ending | Extra trailing statements | 4 |
| | Parentheses / Brackets | Missing closing parenthesis | 5 |
| | Alias / AS | Redundant AS | 3 |
| | SELECT List | Missing column list after SELECT | 1 |
| Identifiers & Objects | Variables/Placeholders | Variable error | 13 |
| | | Missing partition conditions in DELETE statement | 2 |
| | | Partition column comparison with numeric type not allowed | 2 |
| | Ambiguous References | Column exists in multiple tables but alias omitted | 8 |
| | | Ambiguous alias in nested subquery with same column name | 1 |
| | Schema/Object | Field/Table does not exist | 11 |
| | | Missing partition query conditions | 2 |
| | Naming/Alias | Duplicate names (column/alias) | 5 |
| Joins & Grouping | GROUP BY | Missing grouping column | 14 |
| | | Missing HAVING clause for aggregate filtering | 1 |
| | JOIN Ambiguity | Missing condition causing Cartesian product | 6 |
| | | Missing table prefix for duplicate column names in join | 35 |
| | Nested Joins | Ambiguous column reference due to multiple levels of alias | 1 |
| Punctuation & Formatting | Punctuation/Parentheses | Punctuation error | 49 |
| | | Incorrect quote type for column alias with special characters | 4 |
| | | Missing semicolon between statements | 5 |
| DML & DDL | Insert Statement | Insert error | 37 |
| | | Mismatched column count | 5 |
| | Create Table Statement | Table creation error | 10 |
| Compatibility/Dialect | Function Differences | TRANSFORM with lambda expression not supported in Hive | 3 |
| | | wm_concat function not supported in the current SQL dialect | 1 |
| Types & Data Formats | Type System | Type mismatch | 16 |
| | Date/Time | to_unix_timestap typo | 2 |

org.apache.calcite.runtime.CalciteContextException:: from line 139, column 37 to line 139, column 48: Column 'passenger_id' not found in table 'result_base'

Figure 12. The example of Squirrel-Syntax, where an explicit error message exists.

In the original table, **passenger_id** is not a unique key. Could you help me check why the output contains a large number of duplicate rows? Please fix the bug.

*Figure 13.* The example of Squirrel-Semantic.

