# OpenReview forum: "Beyond Text-to-SQL: Can LLMs Really Debug Enterprise ETL SQL?"
_ICML.cc/2026/Conference — ICML 2026 regular_

### Official Review · Reviewer_eGiv · 2026-03-09

**Soundness:** 3
**Presentation:** 3
**Significance:** 3
**Originality:** 2
**Overall Recommendation:** 4
**Confidence:** 4

**Summary:**

The authors investigate the challenge of evaluating LLMs on enterprise-scale SQL debugging rather than the more common text-to-SQL generation task. The paper introduces the Squirrel Benchmark, which consists of long SQL scripts with injected bugs designed to simulate enterprise ETL workflows. The scripts are significantly larger than typical text-to-SQL datasets (around 140 lines).

**Compliance With Llm Reviewing Policy:**

Affirmed.

**Final Justification:**

While they have addressed many of the points I raised, the execution-free setting remains a concern for real-world usage, and since that is a major element in enterprise ETL SQL, I will keep my score.

**Key Questions For Authors:**

1) Since many queries are generated or expanded using LLMs, how exactly have you evaluated that they are representative of real enterprise ETL pipelines rather than synthetic SQL patterns?
2) The evaluation avoids executing queries. Do the proposed metrics correlate with actual execution correctness?

**Limitations:**

Yes

**Strengths And Weaknesses:**

Strengths:
1) The shift from SQL generation to SQL debugging makes sense. In practice people spend more time fixing existing queries than writing them from scratch.
2) The SQL scripts are much larger than typical benchmarks like Spider or BIRD. The paper reports around 140 lines per query with deep nesting, which is closer to real ETL pipelines.
3) The paper evaluates a large set of models and the results are interesting.

Weaknesses:
1) The novelty claim feels somewhat overstated. There has already been work on SQL debugging benchmarks such as BIRD-Critic and work on LLM-based debugging more broadly. The main difference here seems to be query length and scale rather than a completely new task.
2) A large part of the dataset is generated with LLMs (Claude-4-Sonnet) and then expanded into new scenarios. It is unclear how close these scripts are to real enterprise SQL pipelines.
3) The bug injection process mostly makes minimal edits to otherwise correct queries. In practice many SQL bugs are not that localized and involve mistakes across several parts of a query.

---

> ### Author Rebuttal · Authors · 2026-03-27
>
> We thank the reviewer for the thoughtful and constructive feedback. We are encouraged that the reviewer finds the problem setting, scale, and empirical results meaningful. We address the concerns below.
>
> ---
>
> **Concern 1: Novelty Beyond Prior SQL Debugging Benchmarks**
>
> The key novelty lies not in redefining the debugging task itself, but in **reframing it within realistic enterprise constraints.**
>
> **(1) Problem Setting: Enterprise-scale debugging**
>
> We'd like to emphasize that Squirrel is **not a straightforward scaling of prior SQL debugging benchmarks (e.g., BIRD-Critic)**, but rather introduces a **qualitatively different problem regime**. Prior benchmarks operate on short, self-contained SQL snippets (<10 lines), where debugging largely reduces to local pattern matching or token-level repair. In contrast, Squirrel targets enterprise-scale ETL pipelines (140+ lines, deeply nested, multi-stage transformations). This shift introduces fundamentally new challenges:
>
> - Long-context reasoning across dozens of clauses,
> - Cross-stage dependency tracking (e.g., CTE chains, subqueries, aggregations),
> - Global consistency constraints across the query.
>
> These aspects are not captured by existing benchmarks and cannot be recovered by simply increasing SQL length.
>
> **(2) Benchmark Construction Paradigm: Reverse-engineered pipeline**
>
> We propose a reverse-engineering-based construction pipeline grounded in real SQL structures and a hierarchical bug taxonomy derived from production logs. This enables scalable yet structured generation of debugging tasks.
>
> **(3) Practical Evaluation Paradigm: Execution-free evaluation**
>
> We introduce an execution-free evaluation framework (EM, GM, MB) tailored for enterprise settings. To our knowledge, this is the first systematic attempt to evaluate SQL debugging without relying on execution.
>
> **(4) Extensive study revealing new insights into LLM limitations on multi-step debugging**
>
> Experiments show that models that perform strongly on prior benchmarks degrade significantly in this setting, further supporting that this is **not a trivial extension** (e.g., OmniSQL: 87.6% on Spider → 6.4% on Squirrel).
>
> **(5) Our SFT and agent baselines serve as the first systematic baselines for this new task.**
>
> ---
>
> **Concern 2: Realism of LLM-assisted Data Construction**
>
> We'd like to clarify that characterizing Squirrel as LLM-generated is incomplete. Our pipeline is **anchored in real enterprise SQL at every stage**:
>
> - Seed queries are drawn from a curated corpus of 1,000+ real-world SQL scripts, filtered for structural complexity.
> - The bug taxonomy is derived from 268 production error cases, ensuring that injected bugs reflect real failure modes.
> - Scenario expansion is structure-preserving, maintaining AST-level patterns and operator distributions.
> - All ground-truth queries are execution-validated before bug injection.
>
> This design reflects a practical constraint: **real enterprise SQL pipelines are rarely shareable due to privacy and compliance restrictions**. Under this constraint, Squirrel adopts a **semi-synthetic but structure-preserving approach**, which we argue is currently the only scalable way to approximate real-world workloads.
>
> ---
>
> **Concern 3: Minimal-Change Bug Injection**
>
> We appreciate the reviewer’s observation that real-world SQL bugs can be non-local. However, we'd like to clarify that **minimal-change injection is a deliberate and necessary design choice**, rather than a limitation.
>
> Specifically, it enables:
>
> - **Controlled evaluation of localization and repair ability**,
> - **Clear attribution of model errors**, avoiding confounding factors.
>
> Importantly, even with localized edits, the resulting task is **not local in difficulty** due to:
>
> - long-range dependencies across the query,
> - interactions with schema and intermediate results,
> - cascading effects of small errors in large pipelines.
>
> We view Squirrel as establishing a **controlled and reproducible testbed**, complementary to more entangled real-world scenarios.
>
> ---
>
> **Concern 4: Execution-free Evaluation and Its Validity**
>
> While execution-based evaluation is ideal, it is often impractical in enterprise settings. This motivates our execution-free evaluation framework, which we position as a **practical proxy rather than a replacement**.
>
> We validated the reliability of our proposed metrics, particularly the Graph Match (GM) score in Appendix D.1.4:
>
> - **Correlation with Human Judgment:**  We conducted a human evaluation of 300 samples and found that the GM score agrees with expert judgments in over 68% of cases.
> - **Consistency in Model Ranking:** Our analysis shows that the ranking of models by GM score is consistent with the ranking derived from human evaluation.
>
> ---
>
> Overall, we believe Squirrel introduces a new evaluation setting that better reflects real-world enterprise SQL debugging, and we will revise the paper to clarify these contributions and address the reviewer’s concerns.

---

> > ### Author Rebuttal · Reviewer_eGiv · 2026-04-03
> >
> > I thank the authors for their rebuttal. While they have addressed many of the points I raised, the execution-free setting remains a concern for real-world usage, and since that is a major element in enterprise ETL SQL, I will keep my score.

---

### Official Review · Reviewer_JpfQ · 2026-03-10

**Soundness:** 4
**Presentation:** 3
**Significance:** 3
**Originality:** 3
**Overall Recommendation:** 5
**Confidence:** 4

**Summary:**

This paper focuses on building the first enterprise-grade SQL benchmark, addressing the critical limitation of insufficient complexity in existing SQL benchmarks. The authors mathematically formalize the problem formulation and benchmark construction pipeline, and conduct rigorous and comprehensive validation in the experimental section. Throughout the benchmark construction process, the scale and complexity of the dataset are notably elevated, making the benchmark far more closely aligned with real-world enterprise deployment scenarios.

**Compliance With Llm Reviewing Policy:**

Affirmed.

**Final Justification:**

The authors have addressed the issues I was concerned about. However, I still maintain my score: 5/6.

**Key Questions For Authors:**

1.Please explicitly clarify the consistent rationale for the emphasis (including bold highlighting and underlining) applied to specific data points across all tables.
2.The paper only briefly states that raw data is sourced from real-world enterprise applications. Could you please provide specific examples to illustrate the exact data sources and standardized collection methodologies?
3.All results in the paper are presented separately for the syntax and semantic evaluation tasks. What would the end-to-end model performance be if these two components were integrated in the experiments?

**Limitations:**

yes

**Strengths And Weaknesses:**

Existing SQL benchmarks widely suffer from two core flaws: insufficient task complexity, and a substantial misalignment with real-world enterprise usage scenarios. The proposed Squirrel Benchmark leverages multiple strategies to enhance dataset complexity, achieving a much closer fit to real enterprise workloads. The authors design comprehensive experiments for validation, and the results strongly support the authors’ core claims. The paper has a clear, coherent narrative structure and well-defined positioning within the existing literature.
That said, there are minor issues with the formatting and presentation of tables, alongside gaps in methodological transparency. For example, what is the rationale for the bold highlighting in Table 1, and what is the justification for applying this formatting to two data points within the same column? In Table 2, bold highlighting is consistently applied across different modules, yet underlining for the second-ranked result is only present for one module; this formatting requires full standardization across all modules. In addition, regarding the benchmark construction process, the paper only briefly mentions that high-quality raw data is curated from real-world enterprise applications, with no detailed elaboration on this critical step.

---

> ### Author Rebuttal · Authors · 2026-03-27
>
> We sincerely thank Reviewer JpfQ for recognizing the critical need for our enterprise-grade SQL benchmark. We appreciate your valuable feedback and the opportunity to clarify these concerns.
>
> ---
>
> **Concern 1: Data Sources and Collection Methodology**
>
> We apologize for the insufficient detail regarding the data source. As noted in the original manuscript (**Section 3.1 and Appendix C**), the seed enterprise SQL scripts were curated from our internal data platform—a large-scale production environment that supports tens of thousands of developers daily. To clarify this critical step, we will expand the description in the final version with the following specifics:
>
> - **Source:** The scripts are drawn from anonymized production logs of a high-volume data platform, where they are used in real-world ETL pipelines and scheduled analytical jobs.
> - **Selection Criteria:** We applied a multi-stage filtering process to ensure quality and representativeness:
>   - **Representativeness filtering:** We selected scripts based on a composite complexity metric $C(q)=α(D_{AST}(q)+W_{AST}(q))+βL(q)$, retaining only those above a threshold $τ$ to ensure non-trivial structure.
>   - **Correctness validation:** We verified that each selected script executed successfully in production and further validated them using the TQS tool—an enterprise-grade static analyzer based on Apache Calcite—to confirm syntactic validity and schema-level semantics.
> - **De-identification:** All scripts were fully de-identified before processing to remove any proprietary or sensitive information.
>
> ---
>
> **Concern 2: End-to-End Model Performance**
>
> To ensure an objective, automated, and rigorous evaluation, our benchmark intentionally decouples syntax-level and semantic-level tasks.
>
> - Squirrel-Syntax focuses on fixing non-executable SQL scripts with error messages as context.
> - Squirrel-Semantic focuses on executable but semantically incorrect SQL scripts based on natural-language requirements.
>
> This controlled decomposition enables fine-grained diagnosis of model capabilities and allows us to attribute failures to specific reasoning deficits rather than conflating multiple error sources.
>
> We agree that evaluating end-to-end performance is important. In a realistic deployment pipeline, models would first resolve syntax errors and subsequently perform semantic refinement. Incorporating such a sequential evaluation setting is a valuable direction, which we leave for future work.
>
> ---
>
> **Concern 3: Formatting Rationale and Standardization**
>
> We thank the reviewer for pointing out the formatting inconsistency. Our intended convention throughout the manuscript is to denote the best-performing results in bold and the second-best results with underlining. The inconsistency observed in Tables 1 and 2 was due to a typographical oversight. We will carefully revise these tables and ensure that this formatting convention is applied consistently across all modules in the final version of the manuscript.
>
> ---
>
> We are grateful again for the reviewer’s insightful comments, which will help us improve the clarity, consistency, and completeness of our paper. We will incorporate all the suggested revisions into the final version.

---

> > ### Author Rebuttal · Reviewer_JpfQ · 2026-04-03
> >
> > All my concerns have been adequately addressed. The authors have provided detailed responses and made appropriate revisions to the manuscript. I appreciate their careful and thorough work.
> > The paper presents solid and well-executed work. I maintain my original score of 5/6, which indicates a strong accept.

---

> > > ### Author Response · Authors · 2026-04-03
> > >
> > > Thank you for your positive and constructive comments and for recognizing the quality of our work. We highly appreciate your careful review and valuable feedback. We will carefully revise our manuscripts based on your comments.

---

### Official Review · Reviewer_tKr3 · 2026-03-13

**Soundness:** 2
**Presentation:** 2
**Significance:** 3
**Originality:** 3
**Overall Recommendation:** 4
**Confidence:** 3

**Summary:**

This paper tackles the pragmatic yet underexplored domain of enterprise-level SQL debugging, shifting the research paradigm from simplistic Text-to-SQL tasks to complex, large-scale ETL scripts. The work introduces the Squirrel Benchmark, a comprehensive suite comprising hundreds of syntax-error and semantic-error tasks. To circumvent data privacy constraints and prohibitive annotation costs, a novel reverse-engineering pipeline is developed for data construction. Furthermore, the study establishes an execution-free evaluation framework which incorporates exact match, graph match, and modify better metrics to conduct an extensive assessment of ~30 LLMs alongside various SFT and agentic baselines.

**Compliance With Llm Reviewing Policy:**

Affirmed.

**Final Justification:**

The authors have addressed my concerns partially. Although I am not fully convinced, I agree that this is a solid piece of work, therefore I have raised my score to 4.

**Key Questions For Authors:**

See weaknesses.

**Limitations:**

yes

**Strengths And Weaknesses:**

### Strengths
- The Squirrel Benchmark addresses a critical industry gap by shifting from simple Text-to-SQL tasks to complex, enterprise-level ETL scripts.
- The paper introduces an automated pipeline utilizing reverse engineering and AST-based bug injection. This scalable approach enables the generation of high-quality data from production code while mitigating leakage risks, offering an elegant and practical methodology for privacy-conscious dataset curation.
- The evaluation is comprehensive and extensive. It inculdes nearly 30 LLMs alongside various fine-tuning and agentic strategies.
### Weaknesses
- Although the authors emphasize that enterprise-level SQL debugging is an inherently iterative process requiring continuous tool feedback, a significant disconnect persists between this motivation and their experimental methodology. The proposed Squirrel Benchmark evaluates LLMs almost exclusively on one-shot setting. By requiring models to generate a flawless fix immediately upon receiving the error message and schema, the benchmark fails to capture morden LLMs' ability to resolve bugs through successive attempts and environment feedback. Furthermore, the agent-based experiment is under an execution-free setup, which lackes actual data feedback. Ultimately, relying on a single-turn evaluation to address an inherently multi-turn problem severely undermines the paper’s core premise.
- Another concern of this benchmark lies in its pronounced generator bias and its exclusive reliance on rigid automated metrics. The paper explicitly states that all synthetic data was generated by Claude-4. Consistently in Table 2, Claude-4 significantly outperforms all other evaluated models. This strongly suggests a self-preference bias, where a model inherently excels at understanding and repairing its own stylistic code patterns.
- This paper contains several typographical errors that should be corrected, such as:
  - Table 1, the benchmark name is misspelled as "Squrriel-Syntax" and "Squrriel-Semantic" instead of "Squirrel".
  - Figure 2, the "Validation & Analysis" module contains a typo where "esay samples" is written instead of "easy samples."

---

> ### Author Rebuttal · Authors · 2026-03-27
>
> Thank you for the thorough review and for recognizing our contribution to bridging the gap between simple Text-to-SQL and complex enterprise-level SQL debugging. We appreciate your valuable feedback and the opportunity to clarify your concerns.
>
> ---
>
> **Concern 1: One-shot Evaluation vs. Iterative Debugging**
>
> We agree that enterprise SQL debugging is inherently iterative. Below, we clarify how Squirrel accommodates this nature.
>
> **(1) Benchmark scope: evaluation, not simulation.**
>
> **Squirrel Benchmark is an evaluation benchmark rather than an interactive debugging environment.** Its role is to provide a standardized, reproducible testbed for assessing the final problem-solving capability of different methods, which is consistent with established SWE benchmarks.
>
> **(2) One-shot results establish a challenging baseline.**
>
> Table 2 shows that even state-of-the-art LLMs fall far short of real-world deployment standards (e.g., Claude-4 at ~36% GM). These results establish a challenging lower-bound baseline, underscoring the substantial room for improvement and the pressing need for continued research in SQL debugging.
>
> **(3) The benchmark fully supports iterative evaluation.**
>
> We introduced **an agent-based method (Fig 5, Appendix D.3.2) precisely to address the iterative nature of debugging.** This agent operates in a multi-turn loop with feedback.  Our results demonstrate that such iterative methods can significantly boost performance (e.g., Kimi-K2 + Qwen-3-Coder improves EM by 65% over the single-model baseline), proving that the Squirrel Benchmark effectively evaluates and differentiates between iterative methods.
>
> **(4) Agent feedback is execution-like and practical.**
>
> The agent receives **lint-error feedback from TQS** (e.g., syntax errors, schema validation, type mismatches)—the same immediate and actionable feedback a human developer gets from a linter or compiler. This pseudo-execution feedback is efficient and practical, and it fully supports the iterative refinement loop we designed.
>
> ---
>
> **Concern 2: Generator Bias**
>
> We selected Claude-4 because it was, at the time of construction, the most capable model for generating high-quality, structurally complex SQL code. Leveraging a strong model to synthesize realistic data was a practical necessity to ensure benchmark quality. Generator bias is indeed a legitimate concern in LLM-synthesized benchmarks, and we have carefully considered it throughout the construction of Squirrel. Several factors collectively mitigate this risk.
>
> **(1) Evaluation metrics are objective and model-agnostic.**
>
> Our evaluation relies on deterministic metrics—Exact Match (EM), Graph Match (GM) via optimized AST comparison, and Modify Better (MB) based on edit distance. None of these involves LLM-as-a-judge or any form of subjective scoring. Therefore, the evaluation process itself does not favor any particular model.
>
> **(2) Data generation is structurally constrained.**
>
> Claude-4 did not generate SQL freely. As detailed in Sec 3.1, it performed structure-preserving transfer from real-world seed SQL, strictly preserving AST depth, width, and logical patterns—ensuring realistic enterprise style, not model-specific idiosyncrasies.
>
> **(3) Empirical results do not support strong bias.**
>
> If bias were pronounced, Claude-4 would achieve near-perfect performance. Table 2 shows only 36.46% GM on Squirrel-Syntax—far from the ceiling. Moreover, other models perform competitively: DeepSeek-V3.1 (30.49% GM) and Doubao-Seed-1.6 (30.92% GM). These results suggest that Claude-4’s advantage is modest and that the benchmark remains challenging for all models.
>
> **(4) A human-constructed validation set confirms benchmark reliability.**
>
> We additionally built a separate test set of 268 real-world SQL bugs (covering both syntax and semantic errors) manually curated from production logs by SQL developers. We evaluated two representative models on both Squirrel and this human-constructed set.
>
> | Model           | Squrriel-Human GM （%） | Squrriel-Syntax GM（%） | Squrriel-Semantic GM（%） |
> | --------------- | ----------------------- | ----------------------- | ------------------------- |
> | DeepSeek-R1     | 20                      | 21.98                   | 22.09                     |
> | Doubao-seed-1.6 | 20.15                   | 30.92                   | 20.93                     |
> | DeepSeek-V3.1   | 19.4                    | 30.49                   | 14.73                     |
> | Kimi-K2         | 20.90                   | 27.72                   | 20.93                     |
>
> As shown in the table, model rankings and relative performance are highly consistent across the two datasets, demonstrating that Squirrel faithfully reflects real-world debugging difficulty and is not biased toward its generator.
>
> ---
>
> **Concern 3: Typographical errors**
>
> We thank the reviewer for careful reading and for pointing out these typographical errors. We will correct them in the final version.

---

> > ### Author Rebuttal · Reviewer_tKr3 · 2026-04-03
> >
> > Thank you for the detailed rebuttal. I appreciate the clarifications of the role of Squirrel and the additional human-constructed validation.
> >
> > However, I believe my major concern remains only partially addressed. The core issue is not whether the benchmark supports iterative methods, but whether its default evaluation protocol faithfully reflects the inherently iterative nature of enterprise SQL debugging. As it stands, the benchmark primarily evaluates models in a one-shot, execution-free setting, which still appears misaligned with the multi-step debugging.

---

> > > ### Author Response · Authors · 2026-04-03
> > >
> > > We thank the reviewer for the clarification and provide the following additional points:
> > >
> > > (1) As acknowledged, **Squirrel already supports iterative debugging across multiple aspects**, including task formulation (Sec. 2.1), evaluation metrics (Sec. 3.5, App. D.1.2), and baselines (App. D.2–D.3). We would like to clarify the intended scope of Squirrel. The benchmark is not designed to fully simulate the entire iterative debugging workflow, but rather to isolate a core sub-problem: context-conditioned SQL repair given localized error signals (e.g., error messages, schema, and TQS feedback). This design is motivated by practical constraints in enterprise settings.
> > >
> > > (2)  **The benchmark does not enforce a one-shot evaluation protocol**. The default setup evaluates the final repaired SQL given issue, schema, and context using EM/GM/MB, but does not constrain whether the solution is obtained via single-turn or multi-turn interaction(Sec 2.1). We will make this explicit in the revision to avoid ambiguity.
> > >
> > > (3) To further eliminate potential misunderstanding, **we will augment Table 2 with additional multi-iteration results (e.g., agent-based method which has access to TQS feedback)**, making the role of iterative approaches more visible in the main results.
> > >
> > > (4) Finally, the primary goal of Squirrel is to reveal the current performance gap of LLMs on enterprise-level SQL debugging, across both single-step and iterative settings. As shown in our results, even with strong models and iterative strategies, there remains substantial room for improvement.
> > >
> > > We thank the reviewer again for the helpful feedback and will revise the paper to better reflect these points.

---

### Decision · Program_Chairs · 2026-04-30

**Decision:**

Accept (regular)

**Comment:**

This paper addresses the practical yet underexplored domain of enterprise-level SQL debugging, shifting the focus from simple Text-to-SQL tasks to the more complex and large-scale challenges of debugging ETL scripts. It introduces the Squirrel Benchmark, a comprehensive suite consisting of hundreds of tasks involving both syntax and semantic errors. To overcome data privacy limitations and high annotation costs, the authors propose a novel reverse-engineering pipeline for data construction. Additionally, the study establishes an execution-free evaluation framework that incorporates metrics like exact match, graph match, and modify-better to extensively evaluate approximately 30 LLMs, as well as various SFT and agentic baselines.